# SRSF5 Regulates the Expression of BQ323636.1 to Modulate Tamoxifen Resistance in ER-Positive Breast Cancer

**DOI:** 10.3390/cancers15082271

**Published:** 2023-04-13

**Authors:** Ho Tsoi, Nicholas Nok-Ching Fung, Ellen P. S. Man, Man-Hong Leung, Chan-Ping You, Wing-Lok Chan, Sum-Yin Chan, Ui-Soon Khoo

**Affiliations:** 1Department of Pathology, School of Clinical Medicine, Li Ka Shing Faculty of Medicine, The University of Hong Kong, Hong Kong SAR, China; tsoiho@hku.hk (H.T.); nickf821@connect.hku.hk (N.N.-C.F.); ellenman@hku.hk (E.P.S.M.); george09@connect.hku.hk (M.-H.L.); u3006037@connect.hku.hk (C.-P.Y.); 2Department of Clinical Oncology, School of Clinical Medicine, Li Ka Shing Faculty of Medicine, The University of Hong Kong, Hong Kong SAR, China; winglok@hku.hk; 3Department of Clinical Oncology, Queen Mary Hospital, Hong Kong SAR, China; ann_chan81@hotmail.com

**Keywords:** breast cancer, BQ323636.1, SRSF5, tamoxifen resistance, novel target, SRPK1

## Abstract

**Simple Summary:**

Overexpression of BQ can confer tamoxifen resistance in ER +ve breast cancer. BQ is a variant of NCOR2 spliced through the exclusion of exon 11. SRSF5 is found to be involved in regulating the splicing of NCOR2. A low expression of SRSF5 enhances the chance of excluding exon 11 from mRNA, leading to the production of mRNA translating for BQ. Therefore, a low expression of SRSF5 can enhance BQ expression and, thus, tamoxifen resistance. SRPK1 can alter the activity of SRSF5 for BQ expression. Inhibition of SRPK1 can reduce TAM resistance by suppressing BQ expression in ER +ve breast cancer. Therefore, targeting the upstream pathway of BQ will be a possible strategy for reversing TAM resistance.

**Abstract:**

About 70% of breast cancer patients are oestrogen receptor-positive (ER +ve). Adjuvant endocrine therapy using tamoxifen (TAM) is an effective approach for preventing local recurrence and metastasis. However, around half of the patients will eventually develop resistance. Overexpression of BQ323636.1 (BQ) is one of the mechanisms that confer TAM resistance. BQ is an alternative splice variant of NCOR2. The inclusion of exon 11 generates mRNA for NCOR2, while the exclusion of exon 11 produces mRNA for BQ. The expression of SRSF5 is low in TAM-resistant breast cancer cells. Modulation of SRSF5 can affect the alternative splicing of NCOR2 to produce BQ. In vitro and in vivo studies confirmed that the knockdown of SRSF5 enhanced BQ expression, and conferred TAM resistance; in contrast, SRSF5 overexpression reduced BQ expression and, thus, reversed TAM resistance. Clinical investigation using a tissue microarray confirmed the inverse correlation of SRSF5 and BQ. Low SRSF5 expression was associated with TAM resistance, local recurrence and metastasis. Survival analyses showed that low SRSF5 expression was associated with poorer prognosis. We showed that SRPK1 can interact with SRSF5 to phosphorylate it. Inhibition of SRPK1 by a small inhibitor, SRPKIN-1, suppressed the phosphorylation of SRSF5. This enhanced the proportion of SRSF5 interacting with exon 11 of NCOR2, reducing the production of BQ mRNA. As expected, SRPKIN-1 reduced TAM resistance. Our study confirms that SRSF5 is essential for BQ expression. Modulating the activity of SRSF5 in ER +ve breast cancer will be a potential approach to combating TAM resistance.

## 1. Introduction

Breast cancer is a heterogeneous disease with distinctive molecular subtypes. Based on the information from WHO, in 2020, there were 2.3 million women diagnosed with breast cancer, with 685,000 deaths globally, making it the world’s most prevalent cancer (https://www.who.int/news-room/fact-sheets/detail/breast-cancer; accessed on 27 June 2022). Many therapeutic options are now available, which are dependent upon the biological subtyping of breast cancers. Cancers that express oestrogen receptor α (ER +ve) and/or the progesterone receptor (PR +ve) are likely to respond to endocrine therapies such as tamoxifen (TAM). About 70% of breast cancers belong to this class. TAM is usually taken orally for 5–10 years, and can reduce the chance of the cancer recurrence by nearly half [1]. ER-driven signalling pathways are the dominant mechanism for driving the development and progression of ER +ve breast cancer. Therefore, suppressing ER-mediated signalling can effectively suppress breast cancer. TAM is an antagonist of ER. Once TAM binds to the ligand-binding domain of ER, it can prevent ER activation and therefore block ER activity.

Although TAM can effectively prevent breast cancer from recurrence and metastasis in ER +ve breast cancer, half of the patients will eventually develop resistance [2]. Endocrine-resistant patients represent up to 25% of all breast cancer patients, presenting a substantial clinical challenge. Possible mechanisms that confer TAM resistance have been intensely studied. These include the hypermethylation of the ER promoter region, leading to the downregulation of ER expression [3], hyperactivation of signalling cascades such as the PI3K/AKT/mTOR signalling pathway [4], the involvement of SOX9 and SOX2 transcription factors [5,6], and MYC [7]. Inhibiting the PI3K/AKT pathway with drugs targeting PI3K, mTOR, or AKT may improve the effectiveness of chemotherapy and endocrine therapy for breast cancer patients [8,9]. However, given the complexity of the pathway, pathway inhibition at any level may elicit compensatory mechanisms, limiting its effectiveness [10].

Alternative splicing (AS) can generate distinct mRNA isoforms from a single gene, and is shown to play a vital role in cancer progression and therapy [11]. We previously identified BQ323636.1 (BQ), a novel splice variant of NCOR2 that induces TAM resistance [12] and generates a monoclonal antibody that recognizes the unique epitope for the BQ protein (US patent no. US 10,823,735) [13]. BQ results in the exclusion of exon 11 of NCOR2, with exons 10 and 12 joined together to produce mRNA for BQ synthesis. We confirmed that BQ could induce TAM resistance through multiple signalling mechanisms such as HIF-1α signalling [14], the IL-6/STAT3 axis [14], and the IL-8/CXCR1 axis [15]. TAM resistance was reduced by reducing the expression of BQ [16]. Therefore, identifying upstream modulators of BQ would be essential to uncover novel mechanisms for TAM resistance in breast cancer.

Most precursor mRNAs are subject to alternative splicing through which gene expression can be regulated and the diversity of the proteome can be increased [17,18]. Splicing factors (SFs) that bind to specific sequences in an exon can regulate the choice of splice sites [19]. These SFs can act as splice enhancers to enhance the inclusion of an exon or as splice silencers which mediate the exclusion of an exon [20]. The serine/arginine-rich splicing factors (SRSF1-12) are a family of SFs that contribute significantly to regulating pre-mRNA alternative splicing [21]. SRSFs have been demonstrated to be associated with cancer [22,23]. SRSF1 overexpression has been shown to cause an oncogenic effect in various cancers [24]. SRSF2 knockdown has been shown to enhance the expression of GCH1-L and of STK39-L, which are the isoforms involved in cell cycle regulation and DNA repair, to cause an oncogenic effect [25]. SRSF3 overexpression has been shown to enhance the expression of the interleukin enhancer-binding factor 3 (ILF3) isoform 1 and isoform 2, which could promote cell proliferation in lung cancer [26]. Overexpression of SRSF5 has been shown to up-regulate the expression of oncogenic MCM2 and MCM4 [27]. The activities of SRSFs are regulated by phosphorylation mediated by several kinases, such as the serine-arginine protein kinases 1, 2, and 3 (SRPK1-3) [28], SRPK1 being the most well-documented. Aberrant SRPK1 expression has been documented in cancers, breast cancer included [29,30]. In addition, it has been reported that SRPK1 could modulate cisplatin resistance [31] and metastasis [32] in breast cancer. Therefore, the SRPK–SRSF regulatory axis should be considered important in cancer development and drug resistance.

TAM resistance remains a significant obstacle that hinders breast cancer treatment. Identifying possible strategies to combat TAM resistance is urgently needed. In this study, we illustrate a novel strategy to achieve this. We confirm that the down-regulation of SRSF5 results in the skipping of exon 11 of NCOR2, which thus favours BQ production, resulting in TAM resistance. Overexpression of SRSF5, on the other hand, reduces the expression of BQ and compromises TAM resistance. We also confirm that the function of SRSF5 is regulated by SRPK1. Inhibiting SRPK1, while decreasing SRSF5 phosphorylation, increases SRSF5 protein interaction with exon 11 of NCOR2. SRSF5 proteins favour the incorporation of exon 11, producing wild-type NCOR2 rather than BQ. A low expression of BQ compromises TAM resistance. Therefore, inhibiting SRPK1 in ER +ve breast cancer could reduce TAM resistance. Our investigation highlights the feasibility of targeting SRPK1 to reverse TAM resistance.

## 2. Materials and Methods

### 2.1. Mammalian Cell Culture, Transfection and Stable Cell Line Establishment

All cell lines were incubated in a tissue culture incubator with 5% CO_2_ at 37 °C. MCF-10A (a non-cancerous breast cell line), MCF-7 (tamoxifen-sensitive) and ZR-75 (tamoxifen-sensitive) were obtained from American Type Culture Collection (ATCC, Manassas, VA, USA). LCC2 is a tamoxifen-resistant cell line derived from MCF-7, while AK47 is a tamoxifen-resistant cell line derived from ZR-75. LCC2 and AK-47 were kindly provided by Professor Robert Clarke from Georgetown University Medical School, Washington, DC, USA. All cell lines were re-authenticated by short tandem repeat profiling [13].These cell lines were used in our previous studies [13,14,15,16]. MCF-10A was maintained in DMEM/F12 (11330032; Thermo Fisher Scientific, Waltham, MA, USA) supplemented with horse serum (5%; 16050122; Thermo Fisher Scientific, Waltham, MA, USA), EGF (20 ng/mL; PHG0313; Thermo Fisher Scientific, Waltham, MA, USA), hydrocortisone (0.5 mg/mL; H-0888; Sigma-Aldrich, St. Louis, MO, USA), cholera toxin (100 ng/mL; C-8052; Sigma-Aldrich, St. Louis, MO, USA), insulin (10 μg/mL; I-1882; Sigma-Aldrich, St. Louis, MO, USA), and P/S (1%; 10378016; Thermo Fisher Scientific, Waltham, MA, USA). MCF-7, ZR-75, LCC2, and AK-47 cells were cultured in DMEM (12100046; Thermo Fisher Scientific, Waltham, MA, USA) supplemented with FBS (10%; 26140079; Thermo Fisher Scientific, Waltham, MA, USA), and P/S (1%; 10378016; Thermo Fisher Scientific, Waltham, MA, USA). Mycoplasma examination was performed to confirm that all cell cultures were mycoplasma-free. The assay was conducted by the Core Facility provided by the Li Ka Shing Faculty of Medicine, The University of Hong Kong, Hong Kong SAR, China.

Lipofectamine 2000 (11668019; Thermo Fisher Scientific, Waltham, MA, USA) and Oligofectamine (12252011; Thermo Fisher Scientific, Waltham, MA, USA) were employed for plasmid and siRNA transfection, respectively, in accordance to the manufacturer’s instructions. An amount of 1 μg/mL of puromycin (A1113803; Thermo Fisher Scientific, Waltham, MA, USA) or 500 μg/mL of G418 (10131027; Thermo Fisher Scientific, Waltham, MA, USA) was employed for stable cell line selection. The selection process took 6 weeks.

### 2.2. Chemicals, siRNA, shRNA and Plasmids

The substances 4-hydroxytamoxifen (TAM; S7827; Selleck Chemicals LLC, Houston, TX, USA), SRPKIN-1 (HY-116856; MedChemExpress LLC; Monmouth Junction, NJ, USA) and TG003 (HY-15338; MedChemExpress LLC; Monmouth Junction, NJ, USA) were purchased and dissolved in DMSO. SRSF3-targeting siRNA (siSRSF3; sc-38338) and non-targeting siRNA (siCtrl; sc-37007) were obtained from Santa Cruz Biotechnology, Dallas, TX, USA. SRSF5 Human shRNA Plasmid Kit (TL309487; OriGene Technologies, Inc., Rockville, MD, USA) was used. SRSF5 Human Tagged ORF Clone (RC207271) was purchased from OriGene Technologies, Inc., Rockville, MD, USA. pCMV6-Entry Mammalian Expression Vector (PS100001; OriGene Technologies, Inc., Rockville, MD, USA) was used for control overexpression.

### 2.3. Real-Time qPCR and PCR-Based Splicing Assay

TRIzol was employed to extract the total RNA from mammalian cells (15596026; ThermoFisher, Waltham, MA, USA). cDNA was synthesised by PrimeScript^TM^ RT Master Mix (RR036; Takara Biomedical Technology Co., Ltd., Changping District, Beijing, China). An amount of 0.5 µg of RNA was used for cDNA synthesis. cDNA was analysed by PCR using the ProFlex PCR system (Thermo Fisher Scientific, Waltham, MA, USA). qPCR was analysed using SYBR Green Master Mix (A25742; ThermoFisher, Waltham, MA, USA). qPCR was performed using the StepOne Real-Time PCR system (Thermo Fisher Scientific, Waltham, MA, USA). The relative gene expression level was determined by the ΔΔCT method with actin as an internal control. GoTaq^®^ G2 DNA Polymerase (M784A; Promega, Madison, WI, USA) was used to detect the expressions of BQ (352 bp) and NCOR2 (443 bp) amplicons with primers forward-splicing NCOR2-BQ and reverse-splicing NCOR2-BQ (refer to Appendix A for the details). The following PCR conditions (35 cycles) were used: 95 °C for 30 s, 54 °C for 15 s and 72 °C for 15 s. Actin was used as the loading control. The DNA sequences of the primers used are as shown in Table 1.

### 2.4. Cell Viability Assay

All experiments were performed in triplicate. A total of 5000 cells were seeded onto 96-well plates, and cell viability was determined by Cell Counting Kit-8 (C0038; Beyotime, Shanghai, China) and an MTT assay (M6494; Thermo Fisher Scientific, Waltham, MA, USA). The microplate reader Infinite F200 (Tecan, Seestrasse, Switzerland) was employed to determine the absorbance at 570 nm. The reference wavelength was 650 nm. For the clonogenic assay, 2000 cells were seeded onto 2-well plates, with 0.01% crystal violet in PBS (C0775; Sigma, St. Louis, MO, USA) being used to stain the cell colonies. A colony was defined to contain an aggregate of more than 50 cells. For the parental cell line MCF-7, since the colonies were easily distinguishable, the clonogenicity (%) of MCF-7 shCtrl, MCF-7 shSRSF5.1 and MCF-7 shSRSF5.2 was determined. On the other hand, for the parental ZR-75 cell line, colonies were not easily distinguished, hence the solvent extraction method was applied to determine the cell viability of ZR-75 shCtrl, ZR-75 shSRSF5.1 and ZR-75 shSRSF5.2. An amount of 500 μL of isopropanol was added to extract the crystal violet to form a purple solution. An amount of 100 μL of the solution was transferred to a 96-well plate. Triplicate samples were used. The absorbance values at 570 nm and 650 nm were determined using the microplate reader Infinite F200 (Tecan, Seestrasse, Switzerland).

### 2.5. Western Blot and Co-Immunoprecipiation

Proteins from the cells were extracted using a RIPA buffer (50 mM Tris-Cl pH7.4, 150 mM NaCl, 1% NP-40, 0.1% SDS, 1% sodium deoxycholate) supplemented with phenylmethanesulfonyl fluoride (PMSF; P7626; Sigma, St. Louis, MO, USA) and PhosSTOP EASYPack tablets (4906837001; Sigma, St. Louis, MO, USA). A protein assay (5000112; Bio-Rad, Hercules, CA, USA) was employed to determine protein concentrations. An amount of 20 μg of proteins was analysed using SDS-PAGE and transferred to PVDF (1620177; Bio-Rad, Hercules, CA, USA). The following primary antibodies were employed: anti-BQ (1:1000; D-12; Versitech Limited, Hong Kong SAR, China), anti-SRSF5 (1:1000; ab67175; Abcam; Cambridge, UK); anti-SRSF3 (1:1000; #51039; Cell Signaling Technology, Inc.; Danvers, MA, USA); anti-NCOR2 (1:1000; #62370; Cell Signaling Technology, Inc.; Danvers, MA, USA); anti-phosphoepitope SR proteins (MABE50; 1:1000; Merck, Rahway, NJ, USA); anti-GAPDH (1:10,000; sc-32233; Santa Cruz Biotechnology, Dallas, TX, USA); and anti-HSP90 (1:4000; #4874; Cell Signaling Technology, Inc.; Danvers, MA, USA). The secondary antibodies used were anti-rabbit HRP (1:5000; P0260; Agilent Dako, Santa Clara, CA, USA) and anti-mouse HRP (1:5000; P0447; Agilent Dako, Santa Clara, CA, USA). The experiments were repeated at least three times, and representative images were shown.

Cells were lysed in 200 μL of a Co-IP buffer (20 mM Tris-Cl, pH 7.4, 150 mM NaCl, 0.5% NP-40, and 10% glycerol) supplemented with phenylmethanesulfonyl fluoride (PMSF; P7626; Sigma, St. Louis, MO, USA) and PhosSTOP EASYPack tablets (4906837001; Sigma, St. Louis, MO, USA). An amount of 10 μL of the lysate was saved (input fraction). An amount of 90 μL of the lysate was incubated with anti-SRPK (1:200; sc-100443; Santa Cruz Biotechnology, Dallas, TX, USA) or anti-mouse IgG (1:200; X0931; Agilent Dako, Santa Clara, CA, USA) at 4 °C overnight. In total, 50 µL of Dynabead Protein A (10002D; Thermo Fisher Scientific, Waltham, MA, USA) was used. The beads were incubated with the immunoprecipitant at 4 °C for 2 h. The loaded beads were incubated in 1 mL of the Co-IP buffer 3 times at room temperature, followed by a 10 min wash each time. A total of 50 µL of the sample buffer (2 × SDS) was used to incubate the loaded beads at 99 °C for 5 min. Candidate proteins were detected by Western blot. HRP-conjugated Protein A (1:4000; 18–160; Sigma-Aldrich, St. Louis, MO, USA) was used as the secondary antibody.

### 2.6. RNA–Protein Interaction

1 × 10^7^ were resuspended in a 200 μL lysis buffer (with 20 mM HEPES, a pH of 7.4, 150 mM NaCl, 5 mM MgCl_2_, 0.5% Nonidet P-40, and a 40-unit RNaseOUT™ Recombinant Ribonuclease Inhibitor; 10777019; ThermoFisher, Waltham, MA, USA), and 10 µg/mL of yeast tRNA (AM7119; ThermoFisher, Waltham, MA, USA). The suspended cells were lysed by sonication (70% power; 30 s on/off cycle for 6 cycles) on ice using ultrasonic cell disruptor Branson Digital Sonifier (SFX 150; Branson Ultrasonics Corp; Brookfield CT, USA) with a microtip (109-122-1066; Branson Ultrasonics Corp; Brookfield, CT, USA). The RNA-protein complex was immunoprecipitated by anti-Myc (1:400; #2276; Cell Signaling Technology, Inc.; Danvers, MA, USA) or anti-rabbit IgG (1:400; X0903; Agilent Dako, Santa Clara, CA, USA) at 4 °C overnight. No antibody control was included. The immunoprecipitant was incubated with 50 μL of Protein A agarose beads (20333; ThermoFisher, Waltham, MA, USA) at 4 °C for 2 h. The beads were incubated with 1 mL of the lysis buffer three times with rotation at 4 °C, followed by 10 min washing each time. An amount of 1 mL of the Trizol reagent was employed to extract associated RNA. RT-qPCR was performed to detect the target mRNA. NCOR2-exon 11-F (5′-AGA AGA AGG TGG AGC GCA T-3′) and NCOR2-exon 11-R (5′-GCT GCT TGC GGA TCT CAG-3′) were used. The sequences were as follows. HSP70-3′-UTR-F: AGG CCA GGT TCT TAG CAC AT; HSP70-3′-UTR-R: GGC CAG TTG TAA AGG GTT GG; β-globin-3′-UTR-F: CTC GCT TTC TTG CTG TCC AA; β-globin-3′-UTR-R: CAA GGC CCT TCA TAA TAT CCC C. ∆CT = CT_elute_ − CT_input_ and ∆∆CT = ∆CT_anti-SRSF5 or IgG_ − ∆CT_no Ab_. Expression = 2^−(∆∆CT)^.

### 2.7. Xenograft

The 1 × 10^7^ cells of ZR-75-shCtrl, ZR-75-shSRSF5, LCC2-OE-Ctrl, LCC2-OE-SRSF5 were implanted into the mammary fatpad of nude mice (BALB/cAnN-nu) at 6 weeks of age. When tumours became palpable, the mice were randomly assigned into treatment and control groups. TAM dissolved in peanut oil or saline was used. Each mouse received TAM (0.5 mg) through subcutaneous injection twice per week. The treatment window was 6 weeks. The mice were administered TAM or saline 12 times. An electronic caliper was employed to measure tumour dimensions (the longest diameter (D) and the shortest parameter (d)). The volume (mm^3^) of tumour was calculated according to the formula D × d^2^/2. The procedures received official approval from the University of Hong Kong Committee on the Use of Live Animals in Teaching and Research (5103-19).

### 2.8. Tissue Microarray and Immunohistochemistry

Approval for the clinical study was obtained from the Institutional Review Board of HKU and HA Hong Kong West Cluster (identification number: HKU/HA HKW IRB, no. UW 08-147). Clinical information about the patients was retrieved from the database of Queen Mary Hospital, Hong Kong SAR. Representative histological sections were reviewed by pathologists. Regions from each of the tumours were selected and used for the construction of the tissue microarray (TMA), totalling 137 cases (Table 2). Each tumour was constructed in triplicate, the average score being taken for each case. TMA slides were deparaffinised with xylene and rehydrated in ethanol. Antigen retrieval involved heating them in the Tris-EDTA buffer at pH 8.0 using a pressure cooker for 10 min. The slides were then immersed in 3% H_2_O_2_ solution, and then rinsed twice with TBS. Incubation in primary monoclonal antibodies, anti-SRSF5 (SRP40) (1:50; RN082PW; Medical and Biological Laboratories Co. Ltd., Tokyo, Japan) or monoclonal BQ323636.1 (1:50; D-12, Versitech Ltd., Hong Kong SAR, China) was carried out at 4 °C for 20 h. The slides were rinsed in TBST and then incubated with secondary anti-rabbit (K4003; Agilent Dako, Santa Clara, CA, USA) or anti-mouse (K4001; Agilent Dako, Santa Clara, CA, USA) antibodies for 30 min at room temperature. A further wash with TBST was performed before chromogen DAB/substrate reagent incubation for 10 min. The images of TMA slides were captured using the Aperio ScanScope system (Leica Biosystems, Wetzlar, Germany) for further evaluation. Scoring was performed by two independent individuals. The H-score was used for SRSF5 and BQ323636.1 nuclear expression, calculated as follows: 1 × % of nuclei stained at low intensity + 2 × % of nuclei stained at moderate intensity + 3 × % of nuclei stained at high intensity. The median of the H-score was the threshold, which was 110 for the BQ nuclear score and 110 for the SRSF5 nuclear score.

### 2.9. Statistical Analysis

All numeric data were recorded, maintained and processed in Excel (Microsoft). Statistical analyses were performed using the internal functions of Prism5 (GraphPad) or SPSS25 (IBM). All results were shown as mean value ± SD obtained from at least three independent experiments in a bar chart format. For in vitro studies and a Students’ *t*-test were used to determine the significance between the two groups. A one-way ANOVA with Bonferroni’s post hoc test was employed for comparisons with multiple groups. For animal experiments, a two-way ANOVA with Bonferroni’s post hoc test was employed to determine the statistical significance between groups. For in vivo studies, a Mann–Whitney U test was employed to determine the significance between the clinical samples. TAM resistance was defined as ER +ve breast cancer patients who had been treated with TAM in the adjuvant setting but who eventually developed local cancer recurrence or metastases. All tests were two-sided. A chi-square test was used for hypothesis testing. Overall and disease-specific survival analyses were performed using Kaplan–Meier estimates followed by a log-rank test. The association between clinical–pathological parameters with nuclear BQ expression and with nuclear SRSF5 expression was determined by univariate and multivariate Cox-regression analyses, with relative risk (RR) and the 95% confidence intervals (CI) given. The Omnibus test in SPSS was employed to test the proportional hazards assumption. *, **, and *** represent *p* < 0.05, *p* < 0.01 and *p* < 0.001, respectively.

## 3. Results

### 3.1. SRSF5 Modulated the Expression of BQ

We previously showed that the overexpression of BQ323636.1 (BQ) is involved in the development of tamoxifen (TAM) resistance in ER +ve breast cancer [13]. Understanding the molecular mechanism through which BQ is overexpressed would help us gain insight into developing a novel strategy to combat TAM resistance. From the PCR-based splicing assay, we confirmed that BQ mRNA was enhanced in the two TAM-resistant cell lines LCC2 and AK-47 compared with their parental cell lines, MCF-7 and ZR-75 (Appendix A). Serine and arginine-rich splicing factors (SRSFs) are known to bind to the exonic splicing enhancers (ESS) located on exons, giving rise to the inclusion of that particular exon in the mature mRNA [33]. Hence, when the ESS is not bound with any SRSF, an exon will be excluded from the mature mRNA. We hypothesized that the down-regulation of particular SRSFs would increase the chance of excluding exon 11 of NCOR2, and thus generate mRNA for BQ synthesis. The TAM-resistant cell lines LCC2 and AK-47 have high endogenous BQ expression. We compared the expression of SRSF1-12 between TAM-resistant LCC2 and AK-47 cell lines of high BQ expression, with TAM-sensitive parental cell lines MCF-7 and ZR-75, which have low BQ expression. Through qPCR, we found that SRSF3 and SRSF5 were consistently down-regulated in the TAM-resistant cell lines (Figure 1A and Appendix A). We confirmed by Western blot that the protein expression of both SRSF3 and SRSF5 was reduced in LCC2 and AK-47 (Figure 1B and Appendix A). We employed shRNA to generate two independent clones (shSRSF5.1 and shSRSF5.2) with a SRSF5 stable knockdown using MCF-7 and ZR-75 cell lines. The knockdown efficiency was confirmed by RT-qPCR (Appendix A). Subsequently, we confirmed that only the downregulation of SRSF5 could modulate the expression of BQ in MCF-7 and ZR-75 (Figure 1C and Appendix A) but not SRSF3 (Appendix A). In addition, we found that the overexpression of SRSF5 could reduce the expression of BQ in LCC2 and AK-47 cell lines (Figure 1D and Appendix A). Our results, therefore, indicate that SRSF5 can modulate BQ expression.

### 3.2. SRSF5 Bound to Exon 11 of NCOR2 to Favour the Synthesis of Wild-Type NCOR2 mRNA

To produce BQ, skipping exon 11 of NCOR2 is necessary, as verified from clinical samples [12]. We proposed that SRSF5 binds to exon 11, mediating the splicing process to include this exon in the resultant mRNA, producing wild-type NCOR2. A low expression of SRSF5 would reduce the chance of exon 11 being incorporated into NCOR2 mRNA; thus, BQ would be produced instead of NCOR2 (Figure 2A). First, we confirmed that SRSF5 could bind to exon 11 of NCOR2 in MCF-7 and ZR-75 by studying RNA–protein complexes with NCOR2 exon 11-F and NCOR2 exon 11-R. The cells were transfected with a pCMV-myc-SRSF5 mammalian expression plasmid, and immunoprecipitation of the SRSF5/RNA complex with an anti-myc antibody was carried out. The presence of exon 11 was determined by qPCR (Figure 2B). Furthermore, we found that the ectopic SRSF5 could not interact with 3′-UTR of HSP70 and β-globin (Appendix A). The results, therefore, indicate that the interaction between SRSF5 and exon 11 of NCOR2 should be specific. Next, we found that the knockdown of SRSF5 could enhance the BQ mRNA level while reducing NCOR2 mRNA in MCF-7 and ZR-75, as revealed by the qPCR (Figure 2C) with the use of BQ targeting primers, BQ-F and BQ-R, and NCOR2 targeting primers, NCOR2-F and NCOR2-R (Table 1). In addition, the splicing assay was performed using the primers NCOR2-BQ-splicing-F and NCOR2-BQ-splicing-R, which could amplify both NCOR2 and BQ (Appendix A). The opposite results were found when SRSF5 was overexpressed in LCC2 and AK-47 (Figure 2D and Appendix A). These findings suggest that SRSF5 expression can modulate the expression of NCOR2 and BQ in breast cancer.

### 3.3. SRSF5 Could Modulate TAM Response

Since BQ overexpression is responsible for TAM resistance [14], we hypothesised that the down-regulation of SRSF5 would induce TAM resistance in TAM-sensitive MCF-7 and ZR-75 cell lines. Indeed, the knockdown of SRSF5 in MCF-7 and ZR-75 made the cells more tolerant to TAM (Figure 3A). Similar results were obtained from a clonogenic assay (Figure 3B). Next, we employed the xenograft model. We found that the knockdown of SRSF5 could make the cells TAM-resistant, as revealed by the tumour volume (Figure 3C). Finally, we found that the expression of BQ in tumour samples from the SRSF5 knockdown group was significantly enhanced (Figure 3D and Appendix A).

Conversely, we also showed that overexpression of SRSF5 in LCC2 could reduce TAM resistance. As expected, the results from the cell viability (Figure 4A) and clonogenic assays (Figure 4B) suggested that SRSF5 overexpression could recover TAM sensitivity. Results from the nude mice model also showed similar results (Figure 4C). The expression of BQ was significantly reduced in tumours with SRSF5 overexpression (Figure 4D and Appendix A). Our findings thus also indicate that SRSF5 expression can modulate the TAM response in ER +ve breast cancer.

### 3.4. Clinical Significance of SRSF5 in ER +ve Breast Cancer

Immunohistochemistry was employed to determine SRSF5 and BQ expression in 137 breast cancer samples. The samples were assigned into low and high nuclear expression groups based on the median value obtained (Figure 5A). The expressions of SRSF5 and BQ were found to be negatively correlated (Spearman’s R value = −0.3387; *p* < 0.001), which is in keeping with our in vitro findings. Previously, we had already confirmed that high BQ expression was associated with TAM resistance [13]. As expected, in ER +ve cases, the low expression of SRSF5 was significantly associated with TAM resistance (*p* = 0.015; Figure 5C) and local cancer recurrence (*p* = 0.024; Figure 5D) and metastasis (*p* = 0.002; Figure 5E) as indicated by a chi-square test. A low expression of SRSF5 was also a poor prognostic factor for both overall (*p* < 0.001; Figure 5F) and disease-specific (*p* = 0.001; Figure 5G) survival, as revealed by a Kaplan–Meier survival analysis with the log-rank test. The univariate cox regression analysis also showed that cases with high SRSF5 expression were significantly associated with both reduced overall (*p* < 0.001; RR = 0.182; 95% CI 0.071, 0.466; Table 3) and disease-specific survival (*p* = 0.003; RR = 0.142; 95% CI 0.039, 0.519; Table 3), which remained significant in the multivariate cox-regression analysis; cases with high SRSF5 expression showed both reduced overall (*p* = 0.041; RR = 0.087; 95% CI 0.008, 0.901; Table 4) and disease-specific survival (*p* = 0.039; RR = 0.178; 95% CI 0.035, 0.920; Table 4). These findings suggest that low SRSF5 expression is a poor prognostic factor in ER +ve breast cancer.

### 3.5. Inhibition of SRPK1 Could Reduce BQ Expression and TAM Resistance by Altering SRSF5 Activity

Given the above findings, we proposed that TAM resistance could be reduced by altering the activity of SRSF5. Cdc2-like kinase 1 (CLK1) has been reported to regulate the activity of SRSF5 [34]. TG003 is an inhibitor of CLK1 [35]. We identified 50 nM of TG003 as the maximum non-lethal dosage in the non-cancerous breast cell line MCF-10A (Appendix A). However, when we treated LCC2 and AK-47 with 50 nM of TG003, it could not reduce the mRNA expression of BQ and NCOR2 based on the results from qPCR (Appendix A). As expected, this treatment could not reduce TAM resistance, as revealed by the cell viability assay (Appendix A). Using the STRING database [36], we found that serine/arginine-rich splicing factor (SRSF) protein kinase-1 (SRPK1) would interact with SRSF5 (Appendix A). By co-immunoprecipitation, we confirmed that SRPK1 could interact with SRSF5 in LCC2 (Figure 6A). Next, we proposed that the inhibition of SRPK1 might alter the splicing activity of SRSF5 and, thus, BQ expression. We employed SRPKIN-1 to target SRPK1 [37]. We confirmed that 100 nM of SRPKIN-1 was the maximum non-lethal dosage on MCF-10A (Appendix A). Next, we found that 100 nM of SRPKIN-1 could reduce BQ expression while enhancing NCOR2 at both mRNA (Figure 6B) and protein (Figure 6C and Appendix A) levels, with SRSF5 protein expression remained the same. Next, we evaluated the effect of SRPKIN-1 on SRSF5 activity at the phosphorylation level. Since no available antibody specifically recognises phosphorylated SRSF5, we needed to perform an indirect method using co-immunoprecipitation and an anti-phosphoepitope SR protein antibody. This antibody can detect phosphorylated SRSF proteins but is not specific to SRSF5. Hence, immunoprecipitation was needed to isolate the Myc-tagged SRSF5 and thus detect the degree of SRSF5 phosphorylation. The results showed that SRPKIN-1 could reduce the degree of phosphorylation on SRSF5 (Figure 6D and Appendix A).

Reduced SRSF5 phosphorylation resulted in the proportion of SRSF5 interacting with exon 11 of NCOR2 to be increased (Figure 6E), in keeping with the decreased BQ and increased NCOR2, mRNA and protein expression demonstrated on SRPK1 treatment (Figure 6B,C). As our earlier findings had shown, increased binding of SRSF5 to exon 11 would result in preferential NCOR2 expression over BQ expression, leading to decreased TAM resistance. In keeping with this, we also confirmed that treatment with SRPKIN-1 could reverse TAM resistance (Figure 6F). Our findings, therefore, indicate that modulation of SRSF5 activity would be a potential approach to reduce the expression of BQ and, thus, reverse TAM resistance.

## 4. Discussion

Tamoxifen (TAM) resistance is an obstacle to adjuvant therapy for breast cancer treatment. Although ER +ve breast cancer patients initially show a positive response to TAM, up to 50% of patients eventually develop resistance [10]. Loss of the ER during the treatment window is one of the reasons for not responding to TAM. It has been demonstrated that the CpG island of the ER promoter region is hypermethylated in TAM-resistant breast cancer cells [38]. In addition to DNA methylation, histone modification and miRNA have been implicated in TAM resistance [39,40]. Therefore, epigenetics contribute significantly to the development of TAM resistance. However, it would be less feasible to develop a therapy to combat TAM resistance through interfering with DNA methylation, histone modification or miRNA as this would systematically impact healthy cells. Cytotoxicity cannot be neglected.

Receptor-tyrosine kinases are a class of enzyme-linked receptors, which include the epidermal growth factor receptor, insulin and insulin-like growth factor-1 (IGF1) receptor, vascular endothelial growth factor receptor, etc., which have been found to confer tamoxifen resistance [4]. HER2, an EGF receptor, can activate the PI3K/AKT signalling pathway, which is one of the critical mechanisms of TAM resistance [41]. The IGF1 receptor activating PAK2 and VEGF overexpression, which leads to the activation of MAPK/ERK signalling, modulates the development of tamoxifen resistance [42,43,44,45]. These pathways can also crosstalk with each other to make the mechanisms of TAM resistance more complex. Therefore, targeting a single pathway might not effectively combat TAM resistance. However, the cytotoxicity would be too strong to manage when targeting these pathways altogether.

Transcriptome changes have also been shown to contribute to TAM resistance [46]. The transcriptome is governed by transcription factors [47], alternative splicing [48], non-coding RNAs [49] and the epigenome [50]. Alterations in the transcriptome would impact the proteomics [51], with altered protein expression affecting signalling mechanisms. Splicing increases the complexity and diversity of the genome, as a single pre-mRNA can generate various alternative spliced isoforms and thus translate to different protein isoforms [52]. Dysregulation of alternative splicing can lead to aberrant protein isoforms, which may contribute to cancer development [53]. Changing the splicing pattern of the spleen tyrosine kinase gene (SYK) has been shown to impair cell cycle progression [54]. Studies have demonstrated that alternative splicing isoforms could be novel therapeutic candidates for human diseases such as breast cancer [55,56,57,58].

Our previous studies identified a novel splice isoform of NCOR2, called BQ323636.1 (BQ), which is associated with TAM resistance in ER +ve breast cancer [12]. BQ has been shown to modulate various molecular mechanisms, such as HIF-1α-, NF-ĸB-, and AR-driven pathways, to confer TAM resistance [14,15,16]. Knockdown of BQ can compromise TAM resistance, highlighting the importance of BQ in breast cancer, especially in TAM resistance [15]. In this study, we identified that SRSF5 could modulate the alternative splicing of NCOR2 (Figure 1). SRSF5 was found to interact with exon 11 of NCOR2 (Figure 2). The presence of SRSF5 favours the inclusion of the exon 11, resulting in mRNA translating for wild-type NCOR2. Reduced expression of SRSF5 favours exon 11 exclusion, resulting in mRNA translating for BQ. As a functional consequence, TAM resistance would result from low SRSF5 expression. In our xenograft model, we confirmed that the expression of SRSF5 could modulate the TAM response (Figure 3 and Figure 4), indicating the functional significance of SRSF5 in TAM resistance. The negative correlation of SRSF5 expression with BQ overexpression was further examined using the primary cancer samples in the tissue microarray (TMA) to validate this notion. The protein expression of SRSF5 and BQ was determined by immunohistochemistry (IHC). The results confirmed that BQ and SRSF5 were negatively correlated (Figure 5B). As expected, a low expression of SRSF5 was associated with TAM resistance (Figure 5C), local cancer recurrence (Figure 5D) and metastasis (Figure 5E) since ineffective TAM treatment can result in cancer recurrence and metastasis [59]. Our results validated our hypothesis and suggested that low SRSF5 expression should be of clinical significance. Survival analyses confirmed that patients with low SRSF5 expression had poorer outcomes (Figure 5F,G), as also shown by univariate and multivariate cox regression analyses. Conversely, ER +ve breast cancer patients with high SRSF5 expression had a reduced risk ratio (Table 3 and Table 4). These findings confirm that SRSF5 should be involved in modulating NCOR2 production. Reducing SRSF5 expression would favour the production of BQ and induce the development of TAM resistance.

Previously, through a PCR array, we found that BQ overexpression can activate various cancer-related pathways [14]. BQ is an N-terminal part of NCOR2. NCOR2 binds to itself in an anti-parallel fashion, forming a homodimer and giving rise to a functional gene repressor complex [60]. Through mechanistic study, we showed that BQ can interact with NCOR2 to disrupt the repressor activity of NCOR2. In ER +ve breast cancer, NCOR2 represses the activity of ER. However, this repressive activity is compromised in the presence of BQ overexpression. BQ can modulate the interaction of NCOR2 with other transcription factors [14,15,16,61]. Therefore, it would not be surprising that BQ can modulate different mechanisms to confer TAM resistance. Thus, by reducing the expression of BQ, it might be possible to suppress various pathways to overcome TAM resistance. Checkpoint kinase 2 (CHK2) has been shown to regulate the protein stability of BQ via post-translational modification; suppressing CHK2 could reduce the expression of BQ and thus TAM resistance [62]. This study confirms that SRSF5 is an upstream mediator of BQ.

SRSF5 has been shown to modulate apoptosis in lung cancer by controlling the alternative splicing of CCAR1 [63]. Chen et al. [63] demonstrated that acetylation mediated by TIP60 and ubiquitination mediated by SMURF1 could modulate the expression and activity of SRSF5. Cdc2-like kinase 1 (CLK1) has been shown to enhance the phosphorylation of SRSF5 on serine 250 [34]. SRSF5 phosphorylation could modulate the splicing of METTL14 by inhibiting exon skipping while it promoted Cyclin L2 exon skipping [34]. Thus it is also possible that the inhibition of SRSF5 modulators such as CLK1 might alter BQ expression. However, we found that the inhibition of CLK1 was unable to affect BQ expression (Appendix A). Fortunately, SRPK1 was found to interact with SRSF5 (Figure 6A and Appendix A). Inhibition of SRPK1 by SRPKIN-1 could reduce BQ expression but enhance NCOR2 expression (Figure 6B,C). Our results suggest that SRPKIN-1 could reduce the degree of phosphorylation on SRSF5 (Figure 6D) and enhance the proportion of SRSF5 interacting with exon 11 of NCOR2 (Figure 6E). This would favour the production of mRNA for NCOR2 expression. Therefore, as illustrated by Western blot, SRPKIN-1 could reduce the protein level of BQ (Figure 6C). BQ overexpression leads to TAM resistance [12,13]. Therefore, SRPKIN-1 could reverse TAM resistance by inhibiting the exon splicing event, leading to a reduced production of BQ mRNA.

Targeting epigenetic mechanisms or various kinase signalling pathways is not a practical approach to combat TAM resistance in breast cancer. Our work indicates that BQ was differentially expressed to induce TAM resistance through multiple mechanisms [13,14]. Reducing BQ expression could compromise TAM resistance [15,62], highlighting the importance of BQ in breast cancer. Therefore, any approach that can reduce the expression of BQ may be a potential method to reduce TAM resistance. In the current study, we confirmed that SRSF5 is an upstream factor that governs BQ expression. Altering either SRSF5 expression or its phosphorylation could modulate BQ expression and, thus, TAM resistance. These are novel approaches for developing strategies for reducing TAM resistance. As an exploratory study, we have confirmed that SRPKIN-1 could do so. It is possible that other types of post-translational modification of SRSF5 by different modulators can give rise to similar effects. Identifying modulators that would suppress alternative splicing for BQ production would help identify novel targets for combating TAM resistance.

## 5. Conclusions

Our results indicated that BQ expression was governed by SRSF5 expression. SRSF5 could bind to exon 11 of NCOR2 to ensure the inclusion of this exon in the mature mRNA. When SRSF5 was reduced, the chance of excluding exon 11 would be enhanced, favouring the product of mRNA for BQ synthesis. TAM response would be altered by SRSF5 expression. An in vivo study indicated that a low expression of SRSF5 was associated with high BQ expression, TAM resistance, local recurrence and metastasis. Furthermore, ER +ve breast cancer patients with low SRSF5 would have poorer overall and disease-specific survival outcomes.

## 6. Patents

Anti-BQ antibody (D12; US patent no. US 10,823,735).

## Figures and Tables

**Figure 1 cancers-15-02271-f001:**
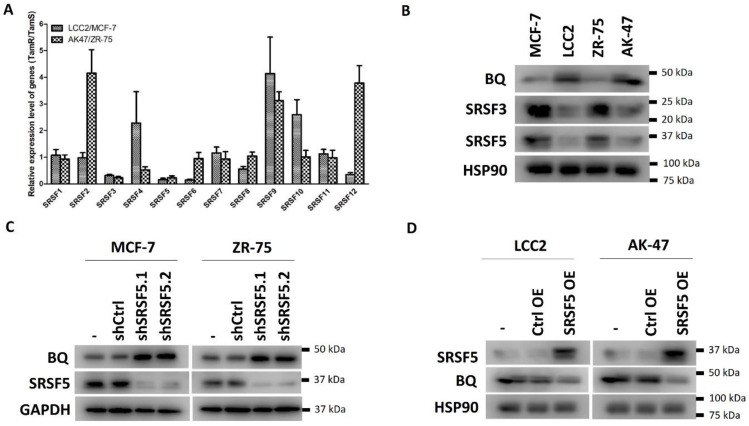
Expression of BQ modulated by SRSF5 in breast cancer. (**A**) SRSF5 and SRSF3 down-regulated in both LCC2 and AK47 cell lines. LCC2 and AK-47 are high BQ-expressing cell lines with tamoxifen (TAM) resistance, while MCF-7 and ZR-75 are TAM-sensitive cell lines with low BQ expression. Abbreviations for TAM-resistant and TAM-sensitive are TamR and TamS, respectively. TamR/TamS indicates the comparison between TamR cell lines (LCC2 and AK-47) and TamS cell lines (MCF-7 and ZR-75). qPCR was used to determine the expression of SRSF1-12. Actin was employed as the internal control. ∆∆CT method was used to evaluate the relative gene expression level. The expression of each SRSF was compared between TAM-resistant and TAM-sensitive cell lines. LCC2 and AK-47 were derived from MCF-7 and ZR-75, respectively. The comparison was made between LCC2 vs. MCF-7 and ZK-47 vs. ZR-75. Results are shown as mean ± SD from three independent experiments. (**B**) Protein expressions of BQ, SRSF3 and SRSF5 determined by Western blot in different cell lines. HSP90 was the loading control. (**C**) Knockdown of SRSF5 enhanced protein expression of BQ in MCF-7 and ZR-75. GAPDH was the loading control. (**D**) Overexpression of SRSF5 reduced BQ expression. LCC2 and AK-47 were stably transfected with pCMV-Myc-SRSF5 (SRSF5 OE) or pCMV (Ctrl OE). The expression of the protein candidates was determined by Western blot with HSP90 or GAPDH as the loading control. Results were obtained from four independent experiments.

**Figure 2 cancers-15-02271-f002:**
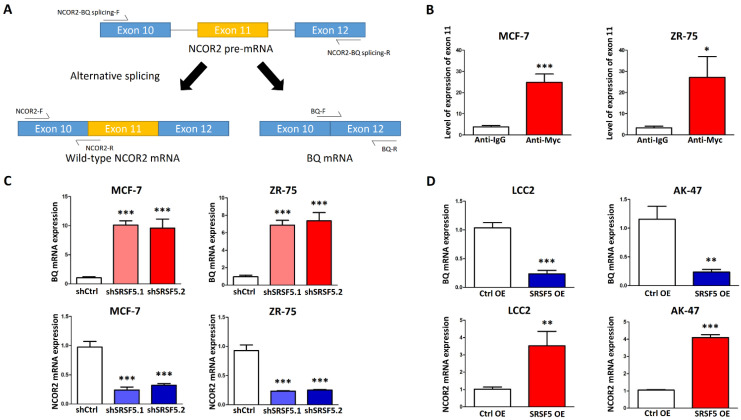
SRSF5 bound to exon 11 of NCOR2 to facilitate the splicing process to generate NCOR2 mRNA. (**A**) Schematic diagram of splicing event that produces NCOR2 and BQ. The arrows indicate the positions of the specific primers used for detecting NCOR2-BQ splicing (NCOR2-BQ splicing -F and -R), NCOR2 and BQ. The diagram is not in scale. (**B**) SRSF5 could bind to exon 11 of NCOR2. MCF-7 and ZR-75 cells were transfected with pCMV-myc-SRSF5 mammalian expression plasmid. The cells were harvested 48 h post-transfection. Anti-myc antibody was employed to immunoprecipitate the SRSF5/RNA complex. The presence of exon 11 was determined by qPCR. The expression level was relative to the input. Results were shown as mean ± SD from three independent experiments. Students’ *t*-test was employed to determine the statistical significance. (**C**) Knockdown of SRSF5 could enhance the expression of BQ while reducing that of NCOR2 in MCF7 and ZR-75 cells. qPCR was employed to determine the mRNA expression of BQ and NCOR2. Actin was the internal control. The bar charts showed results as mean ± SD from three independent experiments. A one-way ANOVA with a Bonferroni post hoc test determined the statistical significance between shCtrl and shSRSF5.1 or shSRSF5.2 groups. (**D**) SRSF5 overexpression could reduce the expression of BQ while enhancing that of NCOR2 in LCC2 and AK-47 cells. BQ and NCOR2 expression levels were determined by qPCR. The results showed the mean expression of BQ and NCOR2 ± SD from three independent experiments. The statistical significance between groups was determined by students’ *t*-test. *, **, and *** indicate *p* < 0.05, *p* < 0.01 and *p* < 0.001, respectively.

**Figure 3 cancers-15-02271-f003:**
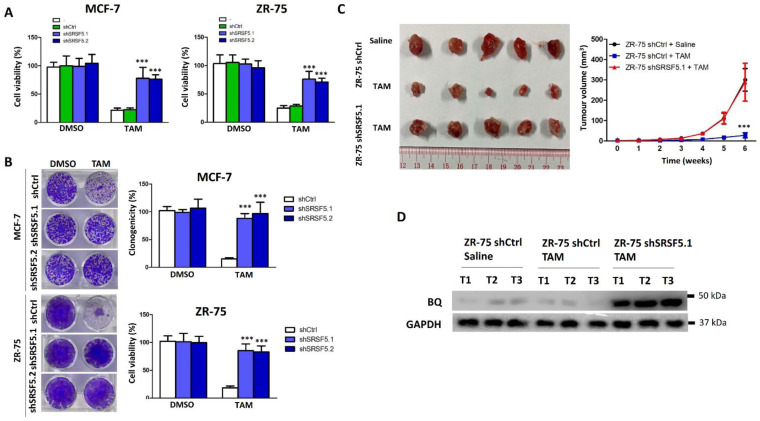
Knockdown of SRSR5 could induce tamoxifen (TAM) resistance. (**A**) Knockdown of SRSF5 could make MCF-7 and ZR-75 become TAM-resistant. The cells were treated with 5 μM of TAM, and a cell viability assay was conducted after 96 h of the treatment. (**B**) Results of the clonogenic assay. The cells were treated with 5 μM of TAM for 21 days. Results are shown as mean ± SD from five independent experiments. A one-way ANOVA with a Bonferroni post hoc test was used to determine the statistical significance between shCtrl and shSRSF5.1 or shSRSF5.2 groups. Clonogenicity (%) was determined in the MCF-7 panel, while cell viability (%) was determined in the ZR-75 panel due to the lack of a countable colony. (**C**) Knockdown of SRSF5 could confer TAM resistance in vivo. ZR-75 shCtrl and ZR-75 shSRSF5.1 cells were engrafted onto the mammary fatpad of nude female mice. TAM or saline was given to the mice via subcutaneous injection. A one-way ANOVA with a Bonferroni post hoc test was used to determine the statistical significance between ZR-75 shCtrl and ZR-75 shSRSF5.1 groups treated with TAM. (**D**) Knockdown of SRSF5 could enhance BQ expression in the tumour tissues. Proteins were extracted from the tumour tissues, and Western blot was employed to determine BQ expression. Protein lysates from 3 independent tumours from each of the groups were analysed. GAPDH was the loading control. *** represents *p* < 0.001.

**Figure 4 cancers-15-02271-f004:**
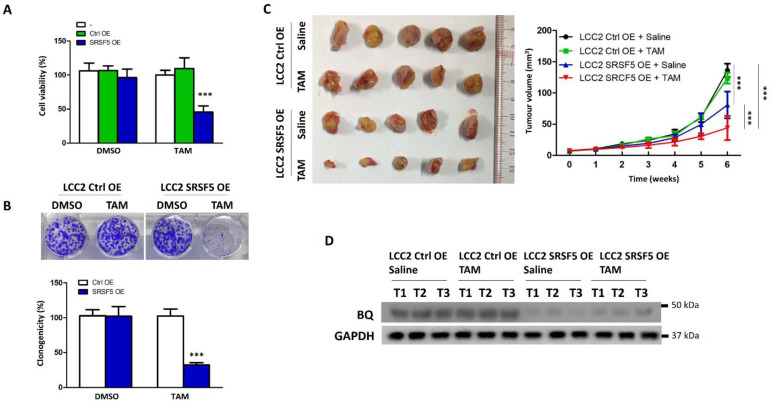
Overexpression of SRSF5 could compromise tamoxifen (TAM) resistance. (**A**) Overexpression of SRSF5 could make LCC2 sensitive to TAM. An amount of 5 μM of TAM was employed to treat the cells. After 96 h of treatment, cell viability was assayed. Results are shown in the bar chart as mean value of cell viability ± SD from five independent experiments. Statistical significance compared with the untreated group was determined using a one-way ANOVA with a Bonferroni post hoc test. (**B**) Clonogenic assay indicated the effect of SRSF5 overexpression on TAM resistance in LCC2. An amount of 5 μM of TAM was employed to treat LCC2 cells for 3 weeks. The bar chart shows the results of mean value of cell viability ± SD from five independent experiments. Statistical significance between Ctrl OE and SRSF5 OE was determined by students’ *t*-test. (**C**) Overexpression of SRSF5 could recover the sensitivity to TAM in vivo. LCC2 Ctrl OE and LCC2 SRSF5 OE cells were engrafted onto the mammary fatpad of female nude mice. TAM or saline was given to the mice via subcutaneous injection. One-way ANOVA with Bonferroni post hoc test was used to determine the statistical significance between indicated groups. (**D**) Overexpression of SRSF5 could reduce BQ expression in the tumour tissues. Proteins were extracted from the tumour tissues, and Western blot was employed to determine BQ expression. Protein lysates from 3 independent tumours from each of the groups were analysed. GAPDH was the loading control. *** represents *p* < 0.001.

**Figure 5 cancers-15-02271-f005:**
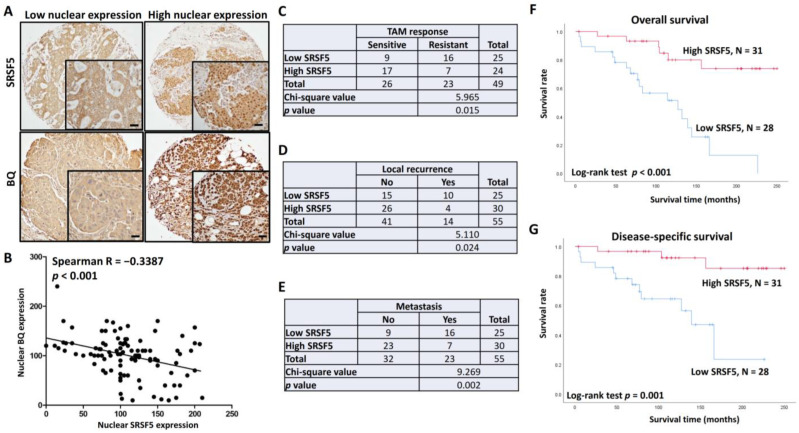
The clinical significance of SRSF5 in ER +ve breast cancer. (**A**) Expressions of SRSF5 and BQ in tumour tissues examined by immunohistochemistry. Representative images are shown. The scale bar represents 20 μm. (**B**) The expressions of SRSF5 and BQ showed a negative correlation. Linear regression was performed to determine the relationship, and the Spearman coefficient was determined. Low expression of SRSF5 was associated with (**C**) tamoxifen (TAM) resistance, (**D**) local recurrence and (**E**) metastasis. Chi-square test was employed. Kaplan–Meier survival analysis showed that the cases of ER +ve breast cancer with low expression of SRSF5 had poorer outcomes of (**F**) overall and (**G**) disease-specific survival.

**Figure 6 cancers-15-02271-f006:**
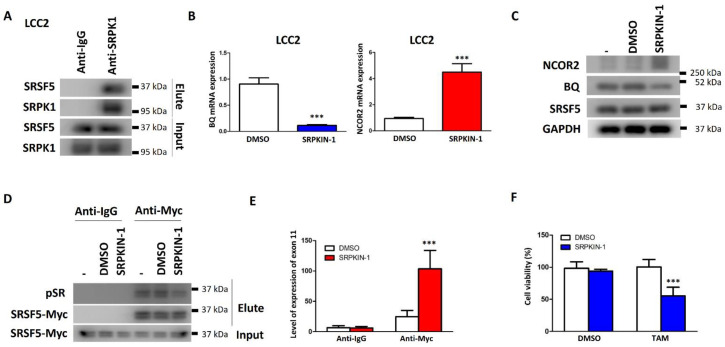
SRPKIN-1 could reverse TAM resistance by modulating the activity of SRSF5 towards exon 11 of NCOR2. (**A**) SRPK1 could interact with SRSF5 in LCC2. SRPK1 was immunoprecipitated. Anti-SRPK1 and anti-SRSF5 were employed to analyse the immunoprecipitant by Western blot. Results were obtained from three independent experiments. SRPKIN-1 could reduce BQ expression but enhance NCOR2 expression at (**B**) mRNA and (**C**) protein levels. qPCR was employed to examine the mRNAs. Actin was the internal control. ∆∆CT method was used to determine the relative expression of the candidates. Results are shown as mean ± SD from three independent experiments. (**D**) SRPKIN-1 could reduce the degree of phosphorylation on SRSF5. LCC2 was transfected with pCMV_Myc_SRSF5. Immunoprecipitation with anti-Myc was performed 48 h post-transfection. pSR represents phosphorylated SRSF proteins. Results were obtained from three independent experiments. *** represents *p* < 0.001. (**E**) SRPKIN-1 could enhance the proportion of SRSF5 interacting with exon 11 of NCOR2. Anti-myc antibody was employed to immunoprecipitate the SRSF5/RNA complex. The presence of exon 11 was determined by qPCR. The expression level was relative to the input. Results are shown as mean ± SD from three independent experiments. Students’ *t*-test was employed to determine the statistical significance between DMSO and SRPKIN-1 groups. (**F**) SRPKIN-1 could reverse TAM resistance. LCC2 was treated with 5 μM of TAM and 100 nM of SRPKIN-1 for 96 h. Cell viability was determined. Results are shown as mean ± SD from four independent experiments. Students’ *t*-test was employed to determine the statistical significance between DMSO and SRPKIN-1 groups. *** represents *p* < 0.001.

**Table 1 cancers-15-02271-t001:** DNA sequences of primers.

	Forward (5′ → 3′)	Reverse (5′ → 3′)
SRSF1	GCG GTC TGA AAA CAG AGT GG	ACA AAC TCC ACG ACA CCA GT
SRSF2	TCC AAG TCC AAG TCC TCG TC	GCG ACC TGG ATT TGG ATT CC
SRSF3	CGA AGT GTG TGG GTT GCT AG	AGT TCC ACT CTT ACA CGG CA
SRSF4	GGA AGG TCG AGG AGA GAG TG	GCT GAC TTT GAT CTG GAG CG
SRSF5	GGT GGA AGA GGT AGA GGA CG	TTT GAG ATC CTG CCA GCT GA
SRSF6	CGT GCT TTG GAC AAA CTG GA	TCG AGA CCT GGA TCT GCT TC
SRSF7	CGA AGA AGA AGC AGG TCA CG	GGA GAT GCT GAC CTT GAC CT
SRSF8	GTC TCA CTC GAA GTC TGG GT	TAG ATG AAG ACC TGG ACC GC
SRSF9	GAG AAG CTG GGG ATG TCT GT	TCC AGT TTA CGC AGG GCA TA
SRSF10	ATT TCT ACA CTC GCC GTC CA	GCC GTC CAC AAA TCC ACT TT
SRSF11	GGG GCT CCT ACT CTT GAT CC	TGA AAC GAG ACC AGC AGC TA
SRSF12	AAT AGG AGG CGG TCA GAC AG	GTT CTT GAC TGC CTT GCT GA
Actin	ATC GTG CGT GAC ATT AAG GAG AAG	AGG AAG GAA GGC TGG AAG AGT G
BQ	GGA GCG CAT GCA GAG AAC C	CTG GCG GTC TTT GTA CAC CT
NCOR2	GGT GGA GCG CAT CGA GAA C	CCC GCT GGC CCA CCC TCT GCA TG
NCOR2-BQ splicing	GAG CAGA AGT TCT GCC AGC GC	TTC TCC CGG AAG GTC TCC TTC

**Table 2 cancers-15-02271-t002:** Clinical information about breast cancer patients.

		Number of Cases	Percentage (%)
ER +ve breast cancer		137	100
Age	<56	65	47.4
	≥56	52.6	52.6
T stage	I, II	41	29.9
	III, IV	10	7.3
Lymph node status	Positive	67	48.9
	Negative	56	40.9
Tumour grade	1, 2	49	35.8
	3	74	54.0
Tumour size	<2 cm	14	10.2
	≥2 cm	79	57.7
ER status	Positive	73	53.3
	Negative	25	18.2
PR status	Positive	49	35.8
	Negative	37	27.0
HER2 receptor status	Positive	35	25.5
	Negative	37	27.0
Triple negative status	Yes	13	9.5
	No	71	51.8

**Table 3 cancers-15-02271-t003:** Univariate cox regression analysis of overall and disease-specific survival in ER +ve breast cancer.

		Overall Survival		Disease-Specific Survival	
Clinical–Pathological Parameters	No. of Cases	RR (95% CI)	*p*-Value	RR (95% CI)	*p*-Value
Age	71	1.736 (0.811, 3.718)	0.155	1.421 (0.564, 3.584)	0.456
T-stage	32	5.897 (1.311, 26.531)	0.021	3.931 (0.714, 21.647)	0.116
Lymph node involvement	65	0.948 (0.423, 2.214)	0.896	1.323 (0.478, 3.662)	0.589
Tumour grade	70	1.336 (0.613, 2.913)	0.466	3.319 (1.092, 10.087)	0.034
Histological type	71	1.342 (0.317, 5.689)	0.690	0.791 (0.182, 3.447)	0.755
PR status	59	0.237 (0.096, 0.582)	0.002	0.111 (0.038, 0.330)	<0.001
HER2 status	49	1.110 (0.427, 2.888)	0.830	1.715 (0.499, 5.893)	0.392
Tumour size	54	1.691 (0.390, 7.327)	0.482	26.528 (0.035, 20173.302)	0.333
Cases with high SRSF5	58	0.182 (0.071, 0.466)	<0.001	0.142 (0.039, 0.519)	0.003
Cases with high BQ	64	7.450 (2.559, 21.692)	<0.001	21.188 (2.802, 160.238)	0.003
Either high SRSF5/low BQ or low SRSF5/high BQ	40	0.122 (0.035, 0.421)	0.001	0.060 (0.008, 0.466)	0.007

**Table 4 cancers-15-02271-t004:** Multivariate cox regression analysis of overall and disease-specific survival in ER +ve breast cancer.

		Overall Survival		Disease-Specific Survival	
Clinical–Pathological Parameters	Cases	RR (95% CI)	*p*-Value	RR (95% CI)	*p*-Value
T-stage	21	12.931 (0.708, 236.235)	0.084	2.583 (0.537, 12.436)	0.237
PR status	21	0.361 (0.053, 2.437)	0.296	0.288 (0.084, 0.986)	0.047
Cases with high SRSF5	21	0.087 (0.008, 0.901)	0.041	0.178 (0.035, 0.920)	0.039

## Data Availability

The materials and raw data will be made available to authorised researchers from the corresponding author upon reasonable request.

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
