# Peer review of "SRSF5 Regulates the Expression of BQ323636.1 to Modulate Tamoxifen Resistance in ER-Positive Breast Cancer"

_cancers, 2023, doi:10.3390/cancers15082271_

Round 1

Reviewer 1 Report

For a number of years, the authors of this article have been studying the role of alternative splicing and the acquired resistance to Tamoxifen in breast cancer.

In this article the authors have further deepened the mechanism by which the splicing factor serine /arginine-rich 5 , SRSF5, regulates the expression of the isoform BQ generated by alternative splicing of NCOR2 mRNA, excluding exon 11, being the mechanism of production of BQ  one of the factor that confers the resistance to Tamoxifen.

Authors have found that the levels of SRSF5 and BQ, are negatively regulated and  correlate with Tamoxifen resistance. This is particularly interesting in Estrogen Receptor positive breast cancer treated by endocrine therapy using Tamoxifen. This negative correlation has been proved by in vitro experiment and further demonstrated in vivo studies with patient’s biopsy.

By using String database authors identified SRPK1 kinase interacting with SRSF5 and by co-immunoprecipitation demonstrated that SRSF5 interacts with SRPK1. Inhibition of SRPKP1 affected BQ expression and enhanced that of NCOR2 in the TAM resistant breast cell lines such as LCC2 and AK47suggesting a possible target to recover TAM resistance in this kind of tumour. 

This work is consistent with previous results obtained and published by authors while adding important data for identifiyng a more targeted therapy that prevents relapse due to acquired TAM resistance in breast cancer.

The body of data are appealing and well shown though deserving some improvement in the figures of WB and legends, which contains results and experiments descriptions while in some cases missing to describe the figures itself.

Major and Minor points

1)       In line 294 of Results authors claim figure S2A  undetectable in supplementary files.

2)       Line 294 of Result: shSRSF5.1 and shSRSF5.2 independent clones with SRSF5 stable knockdown. Which cells were used to generate these clones?

3)       Fig1.A  In the legend of figure, authors should clarify any acronym used in the figure e.g. TamR and TamS; so that it allows readers a quick reading of the figure, while legend helps the understanding.

In B blot of SRSF5 appears too diffuse and it is hard to identify the specific band due to the low quality of W.B. and  lack of molecular standard reference. In alternative, authors can present a better new blot.  Authors are invited to add MW reference for all blots of this paper and to specify in the legend the number of experiments done with the same result.

Blot of GAPDH in B should be ameliorated.

In D authors should better control in LCC2 the GAPDH blot which does not appear to be done on the same membrane as the other blots. In general the blots in D are not aligned,  maybe due to uncorrected acquisition. Then, It would be suggested to improve the entire figure in D.

4)       Legend of Fig 2, in B lines 333-337, the sentence describes the role of SRSF5, the steps of the experiments and the results. These descriptions are more appropriate for the results instead of legends.

5)       Line 376 of the text, the authors claim figure 4D, but D is missing in the figure.

6)       Fig 5. In A, the SRSF5 low nuclear expression (upper panel) corresponds to low BQ expression (lower panel) while SRSF5 high nuclear expression corresponds to high BQ expression. This is the result appreciable with the resolution of BQ immunohistochemistry picture. The problem lies in the immunohistochemistry of BQ, which is too dark to distinguish positive nuclei particularly in SRSF5 low expression sample. The suggestion is to ameliorate the figure or if there had been an improper assembly of panels, please check.

7)       Fig 6C, the blot for NCOR2 is too diffuse, and it is hard to see the band position also because of MW reference lack.

In the legend of D, the authors only describe the steps of the experiment and relative results that are more appropriate for the results section but they missed to specify what pSR is. If pSR is the anti-phospho-epitope SR proteins used to recognize the phosphorylated SRSF5 should be specified in the legend.

8)       In lines 533-534 of the discussion is the sentence missing something?

9)       In Bibliography 32 and 33 are the same publication twice cited.

Author Response

Reviewer 1

Comment 1

In line 294 of Results authors claim figure S2A undetectable in supplementary files.

Response:

Thank you for your comment. There was a typo. This should be Figure S2 rather than FigureS2A in the original file. In the revised manuscript, it is Figure S3 which is the quantification of western blot results shown in Figure 1B. We have made the correction in the text.

In Result 3.1

“Next, by western blot, we confirmed that protein expression of both SRSF3 and SRSF5 was reduced in LCC2 and AK-47 (Figure 1B; Figure S3).”

Comment 2

2)       Line 294 of Result: shSRSF5.1 and shSRSF5.2 independent clones with SRSF5 stable knockdown. Which cells were used to generate these clones?

Response:

Thank you for your question. We employed MCF-7 and ZR-75 cell lines to generate stable cell lines, as shown in Figures S4. For greater clarity, we have added this information to the text.

In result 3.1

 “We employed shRNA to generate two independent clones (shSRSF5.1 and shSRSF5.2) with SRSF5 stable knockdown using MCF-7 and ZR-75 cell lines. The knockdown efficiency was confirmed by RT-qPCR (Figure S4).”

Comment 3

3)       Fig1.A In the legend of figure, authors should clarify any acronym used in the figure e.g. TamR and TamS; so that it allows readers a quick reading of the figure, while legend helps the understanding.

Response:

Thank you very much for this suggestion. This indeed helps us improve the manuscript. The information has been added to the figure legend.

In the legend of figure1:

“Abbreviation for TAM resistant and TAM sensitive are TamR and TamS respectively. TamR/TamS indicates the comparison between TamR cell lines (LCC2 and AK-47) and TamS cell lines (MCF-7 and ZR-75).”

Comment 4

In B blot of SRSF5 appears too diffuse and it is hard to identify the specific band due to the low quality of W.B. and lack of molecular standard reference. In alternative, authors can present a better new blot. Authors are invited to add MW reference for all blots of this paper and to specify in the legend the number of experiments done with the same result. Blot of GAPDH in B should be ameliorated.

Response:

Thank you for your valuable comments. We have revised the figures and added the molecular marker. We have also indicated in the legend the number of experiments done.

In D authors should better control in LCC2 the GAPDH blot which does not appear to be done on the same membrane as the other blots. In general the blots in D are not aligned, maybe due to uncorrected acquisition. Then, it would be suggested to improve the entire figure in D.

Response:

Thank you for your comment. Yes, they were not on the same membrane as the other blots. We usually follow the following procedure: After the determination of protein concentration, we aliquot the proteins (50 μg per tube). We detect each candidate protein on individual membranes because an intact membrane is usually required by journals. The lack of alignment of blots in D was indeed due to uncorrected acquisition. Figure 1D is now revised. All uncropped blots can be found in the file of Supplementary figures, (placed at the end).

Comment 5

Legend of Fig 2, in B lines 333-337, the sentence describes the role of SRSF5, the steps of the experiments and the results. These descriptions are more appropriate for the results instead of legends.

Response:

Thank you for your comment. We have revised the main text and included the information to make readers easier to understand the results.

In manuscript text under Result 3.2

“First, we confirmed that SRSF5 could bind to exon 11 of NCOR2 in MCF-7 and ZR-75 by studying RNA-protein complexes with NCOR2-exon 11-F and NCOR2-exon 11-R. The cells were transfected with pCMV-myc-SRSF5 mammalian expression plasmid, followed by immunoprecipitation of the SRSF5/RNA complex with anti-myc antibody. The presence of exon 11 was determined by qPCR (Figure 2B). Furthermore, we found that the ectopic SRSF5 could not interact with 3’-UTR of HSP70 and β-globin (Figure S9). The results, therefore, indicate that the interaction between SRSF5 and exon 11 of NCOR2 should be specific. Next, we found that the knockdown of SRSF5 could enhance BQ mRNA level while reducing NCOR2 mRNA in MCF-7 and ZR-75 as revealed by qPCR (Figure 2C) with the use of BQ targeting primers, BQ-F, BQ-R, and NCOR2 targeting primers, NCOR2-F and NCOR2-R (Table 1). In addition, the splicing assay was performed using the primers NCOR2-BQ-splicing-F and NCOR2-BQ-splicing-R, which could amplify both NCOR2 and BQ (Figure S10).”

Comment 6

Line 376 of the text, the authors claim figure 4D, but D is missing in the figure.

Response:

Thank you for your comment. The letter D was located near the colony image in the previous vision. We have moved the letter “D” closer to the corresponding figure.

Comment 7

Fig 5. In A, the SRSF5 low nuclear expression (upper panel) corresponds to low BQ expression (lower panel) while SRSF5 high nuclear expression corresponds to high BQ expression. This is the result appreciable with the resolution of BQ immunohistochemistry picture. The problem lies in the immunohistochemistry of BQ, which is too dark to distinguish positive nuclei particularly in SRSF5 low expression sample. The suggestion is to ameliorate the figure or if there had been an improper assembly of panels, please check.

Response:

Thank you for your comments. TMA was employed to analyse the expression of BQ and SRSF5 in the tumour tissues by IHC. The images were previously captured from scanned slides using the AperioScan Scope system. This time, we have retaken new photomicrographs directly from the light microscope. In the revised figure, for the low nuclear expression, the nuclei appear empty or with a purple hue which represents the nuclei counterstaining, compared to the dark brown signals in the high nuclear expressing tumour cells.  The cytoplasmic staining was not included in our analysis because it is nuclear BQ expression (influencing transcription) which is essential for Tamoxifen resistance (1).

Comment 8

  1. i) Fig 6C, the blot for NCOR2 is too diffuse, and it is hard to see the band position also because of MW reference lack.

Response:

Thank you for your comment. NCOR2 is a large protein, 275 kDa. NCOR2 forms a protein complex with other proteins. It is technically challenging to obtain a single band for NCOR2 because 2% of SDS might not be sufficient to denature and linearize NCOR2 completely. In addition, post-translational modification of NCOR2 makes it more difficult to obtain a single band. Therefore, a diffuse pattern will be obtained. MW reference has been added to the figure.

  1. ii) In the legend of D, the authors only describe the steps of the experiment and relative results that are more appropriate for the results section but they missed to specify what pSR is. If pSR is the anti-phospho-epitope SR proteins used to recognise the phosphorylated SRSF5 should be specified in the legend.

Response:

Thank you for your suggestions. pSR represents phosphorylated SRSF proteins. We have added this information to the legend. The antibody is not specific to phosphorylated SRSF5. In the market, anti-p-SRSF5 is lacking. Therefore, we could not detect the degree of SRSF5 phosphorylation directly. Instead, we would need to pull down all SRSF5 proteins and employ anti-p-SR to determine the phosphorylation level. Since the immunoprecipitant should only contain SRSF5, the detection using anti-p-SR should not be affected by other SRSF proteins.

Comment 9

In lines 533-534 of the discussion is the sentence missing something?

Response:

Thank you for your comment. We have revised the sentence.

Comment 10

In Bibliography 32 and 33 are the same publication twice cited.

Response:

Thank you for your comment. We have revised the references.

Reviewer 2 Report

Review of:

SRSF5 regulates the expression of BQ323636.1 to modulate tamoxifen resistance in ER-positive breast cancer

Overall comments

This group has extensively published about a pathologic alternatively spliced isoform of NCOR2, BQ323636.1, which confers Tamoxifen resistance. Here, the authors show that SRPK1 kinase phosphorylates the splicing factor SRSF5, which by binding to NCOR2’s exon 11 causes expression of wt NCOR2 but not of BQ323636.1, which lacks exon 11.

Questions:

What is the protein product of BQ323636.1 when exon 11 is excluded? Does this cause a nonsense mutation?

What epitope does the BQ323636.1-specific antibody bind to? Is this antibody available? I looked up Veritech Ltd. but could not find it.

How does BQ cause TAM resistance? Any new insights since Gong et al 2018?

Why did the authors focus solely on SRSFs. Why didn’t they do a more comprehensive RNAseq study comparing the TAM-resistant with the TAM sensitive cell lines.

The spliceosome contains many factors including many SRSFs. Are the other SRSFs involved in alternative splicing of NCOR2? What is their phosphorylation status?

Is SRPK1 directly phosphorylating SRSF5 on serine 250 like CLK1?

What is the functional consequence of inhibition of SRSF’s phosphorylation? The authors do not show direct evidence that de-phosphorylated SRSF5 does not bind to exon 11 and thus causing exclusion of exon 11? Is this their hypothesis? Please be more specific. See line 564

In the in vivo experiment, I am missing a treatment of the mice with TG003 and SRPKIN-1, to see a direct effect on TAM resistance.

Specific comments

Line 564: “These studies suggest that post-translational modification of SRSF5 by different modulators can give rise to different effects.” This sentence needs to be more specific.

Figure 1A, what are the absolute expression levels relative to GAPDH? This info should be in the supplement to better understand the contributions of the various SRSFs.

Figure 1B-D (raw blots), Western blot bands for SRSF5 in 1B are between 37 and 50kDa while for 1C the band is between 25 and 37. It should be around 31kDa according to the Lit.

Figure2B, why do the authors need to use an overexpressing Myc SRSF5 construct and IP with anti-myc to pull down SRSF5-RNA complex? A direct pulldown with SRSF5 antibody of the endogenous SRSF5 would reflect a more physiologic condition. Overexpressing might just pull down anything. A nonspecific RAN control may be important to make it more convincing.

Misc., What is the source for the inhibitor SRPKIN-1?

Author Response

Reviewer 2

Comment 1

What is the protein product of BQ323636.1 when exon 11 is excluded? Does this cause a nonsense mutation?

Response:

BQ323636.1 is the product generated from NCOR2 because of alternative splicing. Wild-type NCOR2 includes exon 10, exon 11 and exon 12. For the splice variant we are dealing with, exon 11 is excluded, with the resultant mRNA containing exon 10 and exon 12. This mRNA will be translated to produce the resultant protein called BQ323636.1. Due to the exclusion of exon 11, the reading frame of exon 12 will be changed; an effective stop codon will be generated, resulting in a smaller protein.

A nonsense mutation occurs in the genome when DNA change gives rise to a stop codon rather than a codon specifying an amino acid. The production of BQ does not involve the DNA change in the genome, the difference being the information used for the protein translation. Thus, strictly speaking, the resultant stop codon generated from splicing should not be considered a nonsense mutation.  

Comment 2

What epitope does the BQ323636.1-specific antibody bind to? Is this antibody available? I looked up Veritech Ltd. but could not find it.

Response:

Thank you for your question. Information of the antibody can be found on the website of the company.

https://versitech.hku.hk/products/anti-bq-antibody

Information of the epitope can be obtained from our patent (US Patent No. US 10,823,735).

Comment 3

How does BQ cause TAM resistance? Any new insights since Gong et al 2018?

Response:

Thank you for your question. We have subsequently demonstrated that BQ can induce various cancer-related pathways (1). NCOR2, being a co-repressor gene, can suppress the activity of various transcription factors. Our studies show BQ can interact with NCOR2, and when present in excess, can diminish the repressor function of NCOR2. We have since shown that the activities of various transcription factors, such as ER (2), NRF2 (3), HSF4 (1), NF-kB (4), AR (5) are enhanced by BQ overexpression. Hence, BQ overexpression can enhance multiple mechanisms that favour the survival of breast cancer cells even in the presence of therapeutic drugs. This can explain why BQ overexpression can confer drug resistance, as also demonstrated in this revised manuscript. 

In discussion section, we have added mention of specific pathway.

“Our previous studies identified a novel splice isoform of NCOR2, called BQ323636.1 (BQ), associated with TAM resistance in ER +ve breast cancer. BQ has been shown to modulate various molecular mechanisms, such HIF-1α-, NF-ĸB-, and AR-driven pathways, to confer TAM resistance.”

Comment 4

Why did the authors focus solely on SRSFs. Why didn’t they do a more comprehensive RNAseq study comparing the TAM-resistant with the TAM sensitive cell lines.

Response:

Thank you for your question and suggestion. This study was hypothesis-driven. We observed that exon 11 of NCOR2 was excluded, leading to the production of BQ. Splicing factors mediate the splicing process through binding to the exon and recruit the splicesome that includes an exon in the resultant mRNA. The members of SRSF proteins (SRSF1-12) belong to this type of splicing factor. We hypothesised that a low expression of any of the SRSF proteins would associate with a high BQ expression. This is how this study started.

We agree there may be other types of splicing mediators involved in BQ production. We do not believe that the SRSF family is the only protein family that affects BQ synthesis. Indeed, RNA seq will offer more comprehensive study and results. It will be our subsequent investigation. We aim to identify all splicing mediators related to BQ production and determine if these mediators will be clinically significant. 

Comment 5

The spliceosome contains many factors including many SRSFs. Are the other SRSFs involved in alternative splicing of NCOR2? What is their phosphorylation status?

Response:

Thank you for your comments. Yes, you are right. It is highly possible that other factors and their phosphorylation level will involve in modulating the alternative splicing of NCOR2. We are planning to identify these factors. In the current study, we focus on SRSF5 and do not intend to introduce and explore other factors because it is a pilot study to verify our hypothesis. With the successful publication of the current work, we can seek funding to support the continuation of the work.  

Comment 6

Is SRPK1 directly phosphorylating SRSF5 on serine 250 like CLK1?

Response:

Thank you for your question. Based on co-immunoprecipitation, we confirmed that SRPK1 could interact with SRSF5. In addition, based on the results of protein-protein interaction from STRING database, we did not detect other kinases that interact with SRSF5. Therefore, it is less unlikely that SRPK1 would phosphorylate another kinase which would then phosphorylate SRSF5. Therefore, SRPK1 should phosphorylate SRSF5. However, we cannot conclude that serine 250 is the only site for the phosphorylation as there can be other possible sites. We can only tell that serine 250 would be one of the possible sites. In addition, there is no available antibody for detecting the phosphorylation of SRSF5 at a particular site. Therefore, we could not illustrate which amino acid would be the target of SRPK1. To do so, mass spectrometry will be needed. However, we do not have funding that covers the experiments. Therefore, we will not be able to do it unless we obtain funding that can support mass spectrometry.

Comment 7

What is the functional consequence of inhibition of SRSF’s phosphorylation? The authors do not show direct evidence that de-phosphorylated SRSF5 does not bind to exon 11 and thus causing exclusion of exon 11? Is this their hypothesis? Please be more specific. See line 564

Response:

Thank you for your comment. In Figure 6E, we determined the interaction between SRSF5 and exon 11 in the presence of SRPK1 inhibitor SRPKIN-1, which reduced the level of SRSF5 phosphorylation, as illustrated in Figure 6D. Based on the results shown in Figure 6E, SRPKIN-1 treatment could reduce the exon 11 attached to SRSF5. The results suggested that non-phosphorylated SRSF5 reduced the tendency to interact with exon 11.

We have revised the sentence. Thank you for your comment. 

In discussion

“In the current study, we confirmed that SRSF5 is an upstream factor that governs BQ expression. Altering either SRSF5 expression or its phosphorylation could modulate BQ expression and, thus, TAM resistance. These are novel approaches for developing strategies for reducing TAM resistance. As an exploratory study, we have confirmed that SRPKIN-1 could do so. It is possible that other types of post-translational modification of SRSF5 by different modulators can give rise to similar effects.”

Comment 8

In the in vivo experiment, I am missing a treatment of the mice with TG003 and SRPKIN-1, to see a direct effect on TAM resistance.

Response:

Thank you for your comment. It is an excellent suggestion to determine if TG003 and SRPKIN-1 will modulate TAM resistance in vivo. Based on in vitro experiments, SRPKIN-1 could reverse TAM resistance (Figure 6F). The reason for testing TG003 and SRPKIN-1 was to illustrate TAM resistance could be modulated by altering SRSF5 phosphorylation in addition to the protein expression level of SRSF5. These findings illustrated the molecular mechanism that governs SRSF5’s  involvement in TAM resistance. We have focused on illustrating the concept only. We have only been able to obtain approval from the animal ethics committee to perform the experiments using SRSF5 knocking and overexpressing cell lines. However, they will not approve the use of TG003 or SRPKIN-1 in animal experiments unless there is evidence to indicate the potential translational value of these chemicals, for example, the publication of the current study. Therefore, your support will be necessary for our future work.

Comment 9

Line 564: “These studies suggest that post-translational modification of SRSF5 by different modulators can give rise to different effects.” This sentence needs to be more specific.

Response:

Thank you for your comment. We have revised the sentence accordingly.

Comment 10

Figure 1A, what are the absolute expression levels relative to GAPDH? This info should be in the supplement to better understand the contributions of the various SRSFs.

Response:

Thank you for your comment. We have included it in the revised manuscript. The results are shown in Figure S2.

 At the very beginning, we compared the expression of SRSF1-12 in MCF-7, LCC2, ZR-75 and AK-47 with reference to MCF-10A using actin as the internal control. Since we employed ∆∆CT method for all the studies, based on the experimental setting, we were unable to obtain the absolute expression level which requires the use of standard curve. Based on our hypothesis, we speculated that low expression of any of the SRSF in TAM resistance breast cancer cells would be important. As a screening, absolute expression level was not essential at that time. Using a common reference would still allow us to compare SRSF1-12 among the four cell lines and provide sufficient evidence for our purpose. 

Comment 11

Figure 1B-D (raw blots), Western blot bands for SRSF5 in 1B are between 37 and 50kDa while for 1C the band is between 25 and 37. It should be around 31kDa according to the Lit.

Response:

Thank you for your careful inspection. We appreciate it. We made this error while we generated the raw blot image showing the chemiluminescent signal and coloured protein marker. The imager contains two cameras, one for capturing chemiluminescent signal (black-and-white; image 1) and another for the coloured image (image 2). We must merge two images to generate the final image containing the chemiluminescent signal and colour protein marker (please see Appendix figure 1). Usually, we employ the default setting of the imager to generate two images with identical sizes. However, the vertical size of some black-and-white images was reduced by 10-20%. When overlaid, the bands would shift upward. We have repeated the experiments to replace the problematic images.

The calculated molecular weight of SRSF5 is about 31. However, the appearance of its molecular weight on Western blot can be larger. SRSF5 contains SR-rich domain which will reduce the migration rate of the protein in SDS-PAGE. In the antibodies ab67175 (https://www.abcam.com/products/primary-antibodies/srsf5-antibody-ab67175.html) and RN082PW (https://www.mblbio.com/bio/g/dtl/A/index.html?pcd=RN082PW, the target band of SRSF5 is about 37kDa. This is as shown in the repeat blots.

Comment 12

Figure2B, why do the authors need to use an overexpressing Myc SRSF5 construct and IP with anti-myc to pull down SRSF5-RNA complex? A direct pulldown with SRSF5 antibody of the endogenous SRSF5 would reflect a more physiologic condition. Overexpressing might just pull down anything. A nonspecific RAN control may be important to make it more convincing.

Response:

Thank you for your comments. Pull-down endogenous SRSF5 proteins would be excellent. However, the IP efficiency of anti-SRSF5 antibodies (ab67175 and RN082PW) was not good. We employed an overexpression system to increase the chance of capturing SRSF5-RNA complex to enhance the IP efficiency. Based on our experience, the anti-myc antibody would be excellent for IP. Therefore, we employed this approach.

As you suggested, we examined whether SRSF5 would pull down non-specific RNA. We determined if SRSF5 would interact with 3’-UTR of HSP70 and β-globin using MCF-7 and ZR-75 with SRSF5 overexpression. A similar RNA-protein interaction study was performed. 3’-UTR of HSP70 and β-globin were undetectable in the immunoprecipitant. These results indicate that SRSF5 could not interact the 3’-UTR of HSP70 and β-globin. This demonstrated that the interaction between SRSF5 and exon 11 of NCOR2 should be specific. The results are shown in Figure S8.

Comment 13

Misc., What is the source for the inhibitor SRPKIN-1?

Response:

Thank you for your question. SRPKIN-1(HY-116856; MedChemExpress LLC; Monmouth Junction, NJ, USA) is commercially available. The information was shown in method 2.2.

Reviewer 3 Report

This manuscript entitled “SRSF5 regulates the expression of BQ323636.1 to modulate tamoxifen resistance in ER-positive breast cancer” submitted by Ho et al. is dissecting the mechanism of how BQ, a novel splice variant of NCOR2, is overexpressed in ER+ breast cancer. The authors show that SPK1-SRSF5-NCOR2 axis functions in regulating the splicing of NCOR2 transcripts and thus the tamoxifen resistance. The experiment is well-designed, and the subject of this manuscript is related to breast cancer, thus, this paper is relevant to Cancers.

However, some data presentation requires further clarification.

Result 3.1: Authors used RT-PCR to quantify two splicing variants and concluded that BQ is overexpressed in specific cell lines. With the same primers, PCR reactions tend to amplify shorter PCR templates within the same tube. Authors should further check the BQ expression through qPCR with BQ-specific primers.

Figure 1B: Please label the observed protein size beside every western blot result. Based on the “raw blots for figure 1B” provided by the authors, the observed SRSF3 protein size (30-37 kd) didn’t match the commercial Ab datasheet suggested position(20-22 kd). Please explain the background signal differences between 4 lanes.

Figure 1D: Same issue as above, the observed SRSF5 in “raw blots for figure 1D” is around 50 kd while the Ab provider’s data shows around 40 kd. Besides, the SRSF5 band size is not consistent with that of “Raw blots for figure 1B” and “Raw blots for figure 6C”, there are two bands in Figure 1B, but only one clear band in Figure 1D with the same cell line.

Figure S4: It seems the upper panel and lower panel are the same blots but labeled with different names in “raw blots for figure S4”

Figure 3B Regarding the colony information experiment, it is not clear how the authors were able to distinguish individual clones as all colonies appeared to have merged together.

Figure 3C.  Tumors from ZR-75 shSRSF5.1 with Saline treatment should be included as the control for TAM-treated tumors to rule out the influence of SRSF5 knockdown toward tumor growth.

Figure 5A Regarding the IHC staining of BQ in Fig. 5b, there is severe background staining (cytoplasmic staining), making it difficult to identify the changes of BQ in the nucleus. It is suggested that the authors consider replacing the antibody or optimizing the staining conditions.

Result 3.5 As authors indicated in the last: “modulation of SRSF5 activity would be a potential approach to reduce the expression of BQ and, thus, reverse TAM resistance” It would be useful to check the in vivo effect of SRPKIN-1 with Xenograft experiment.   

Discussion: The authors spent too many paragraphs recapitulating this paper's Results section. Please focus more on interpreting and describing the significance of your findings in relation to what was already known about the TAM resistance in ER+ breast cancer and explain any new understanding or insights that emerged as a result of your research.

Author Response

Reviewer 3

Comment 1

Result 3.1: Authors used RT-PCR to quantify two splicing variants and concluded that BQ is overexpressed in specific cell lines. With the same primers, PCR reactions tend to amplify shorter PCR templates within the same tube. Authors should further check the BQ expression through qPCR with BQ-specific primers.

Response:

Thank you for your comments. We did employ qPCR to determine the expression of BQ (Figures 2C and 2D), albeit when SRSF5 was knocked down or overexpressed. To clarify, we have also revised the content of the text to indicate the primers used and the nature of the experiment.

Comment 2

Figure 1B: Please label the observed protein size beside every western blot result. Based on the “raw blots for figure 1B” provided by the authors, the observed SRSF3 protein size (30-37 kd) didn’t match the commercial Ab datasheet suggested position (20-22 kd). Please explain the background signal differences between 4 lanes. Figure 1D: Same issue as above, the observed SRSF5 in “raw blots for figure 1D” is around 50 kd while the Ab provider’s data shows around 40 kd. Besides, the SRSF5 band size is not consistent with that of “Raw blots for figure 1B” and “Raw blots for figure 6C”, there are two bands in Figure 1B, but only one clear band in Figure 1D with the same cell line.

Response:

Thank you very much for your comment. We sincerely appreciate your effort. We made this error while we generated the raw blot image showing the chemiluminescent signal and coloured protein marker. The imager contains two cameras, one for capturing chemiluminescent signal (black-and-white; Image 1) and another for the coloured image (Image 2). We must merge two images to generate the final image containing the chemiluminescent signal and colour protein marker (please see Appendix figure 1). Usually, we employ the default setting of the imager to generate two images with identical sizes. However, the vertical length of some black-and-white images was reduced by 10-20%. When overlaid, the bands would shift and not match the original position. We have repeated the experiments using the old and new samples. We have replaced the problematic images.

The inconsistency might be due to the batch-to-batch variation of the antibody. This study was first commenced in 2016. We have ordered anti-SRSF5 several times. We believe the antibodies were generated from different lots. Also, we employed both 10% and 12.5% SDS-PAGE at different stages in our study. The difference in gel percentage and separation time would result in a difference in the final pattern of the bands. In 12.5% gel, SRSF5 would have two bands, while a single band would be found in 10% gel.

To confirm our results, we have re-run the samples in this publication using the current lot of anti-SRSF5 and with 12.5% gel with consistent patterns obtained, supportive that our results are fine (please see Appendix figure 2).

Comment 3

Figure S4: It seems the upper panel and lower panel are the same blots but labeled with different names in “raw blots for figure S4”

Response:

Thank you for your keen observation. We mistakenly incorporated a wrong picture. Raw blots for figure S4 would really be raw blots for Fig 1C, since S4 is quantification of Fig 1C. Besides, it should show the blots for BQ, SRSF5 and GAPDH. The “raw blots for Figure S4” in question show SRSF3 blots, rather than SRSF5 blots.

The mistaken picture was a slide taken in the middle of image preparation. We were comparing the exposure time. Therefore, they were identical. This has been removed and has replaced it with the correct one. We appreciate you pointing this out this mistake to us.

Comment 4

Figure 3B Regarding the colony information experiment, it is not clear how the authors were able to distinguish individual clones as all colonies appeared to have merged together.

Response:

Thank you for your comment. We apologize for failing to be more explicit. For MCF-7 and LCC2, as colonies could be seen clearly, we could count the number of colonies. Since for ZR-75, the cells could not form clear colonies at the endpoint, we employed an indirect method to determine cell viability. Organic solvent isopropanol was used to extract the pigment from the stained cells and absorbance at 570 nm was measured. The relative cell viability was determined by comparing it with the untreated control. This information has been added to the method section.

In method 2.4

“The solvent extraction method was applied when the colonies were not easily distinguished. 500 μL of isopropanol was added to extract the crystal violet to form a purple solution. 100 μL of the solution was transferred to a 96-well plate. Triplicate samples were used. The absorbance at 570 nm and 650 nm was determined using Microplate reader Infinite F200 (Tecan, Seestrasse, Switzerland).”

Comment 5

Figure 3C. Tumors from ZR-75 shSRSF5.1 with Saline treatment should be included as the control for TAM-treated tumors to rule out the influence of SRSF5 knockdown toward tumor growth.

Response:

Thank you for your comments. We agree this should have been included. Indeed, these findings had been confirmed through in vitro experiments. When we applied for the approval of animal work, the committee mentioned that since this was identical to in vitro experiment and it would not provide new insight. Hence, this part of the experiment was removed.

Comment 6

Figure 5A Regarding the IHC staining of BQ in Fig. 5b, there is severe background staining (cytoplasmic staining), making it difficult to identify the changes of BQ in the nucleus. It is suggested that the authors consider replacing the antibody or optimising the staining conditions.

Response:

Thank you for your comments. We appreciate your concern. The images were previously captured from scanned slides using the AperioScan Scope system. This time, we have retaken new photomicrographs directly from the light microscope. In the revised figure, for the low nuclear expressing tumour cells, the nuclei appear empty or with a purple hue which represents the DAPI counterstaining, compared to the dark brown signals in the high nuclear expressing tumour cells. Cytoplasmic staining was not included in our analysis because it is nuclear BQ expression (influencing transcription) which is essential for TAM resistance (1).

Comment 7

Result 3.5 As authors indicated in the last: “modulation of SRSF5 activity would be a potential approach to reduce the expression of BQ and, thus, reverse TAM resistance” It would be useful to check the in vivo effect of SRPKIN-1 with Xenograft experiment.  

Response:

Thank you for your suggestion. It is an excellent suggestion to determine if SRPKIN-1 will modulate TAM resistance in vivo. Based on in vitro experiments, SRPKIN-1 could reverse TAM resistance (Figure 6F). The animal ethic committee allows us to perform the experiments using SRSF5 knocking and overexpressing cell lines. However, they do not approve the use of SRPKIN-1 in animal experiments unless we have evidence to indicate its potential translational value, for example, the publication of the current study. Therefore, your support will be necessary for our future work to determine if SRPKIN-1 will have a therapeutic effect on breast cancer.

Comment 8

Discussion: The authors spent too many paragraphs recapitulating this paper's Results section. Please focus more on interpreting and describing the significance of your findings in relation to what was already known about the TAM resistance in ER+ breast cancer and explain any new understanding or insights that emerged as a result of your research.

Response:

Thank you for your comments and suggestions. We have revised the discussion to highlight the significance of our work. In addition, we have emphasised that potential approaches for combating Tam resistance may be developed based on the upstream mediator of BQ.

In discussion

" Targeting epigenetic mechanisms or various kinase signalling pathways is not a practical approach to combat TAM resistance in breast cancer. Our work indicates that BQ was differentially expressed to induce TAM resistance through multiple mechanisms [13,32]. Reducing BQ expression could compromise TAM resistance [14,63], highlighting the importance of BQ in breast cancer. Therefore, any approach that can reduce the ex-pression of BQ may be a potential method to reduce TAM resistance. In the current study, we confirmed that SRSF5 is an upstream factor that governs BQ expression. Altering either SRSF5 expression or its phosphorylation could modulate BQ expression and, thus, TAM resistance. These are novel approaches for developing strategies for reducing TAM resistance. As an exploratory study, we have confirmed that SRPKIN-1 could do so. It is possible that other types of post-translational modification of SRSF5 by different modulators can give rise to similar effects. Identifying modulators that would suppress the alter-native splicing for BQ production would help identify novel targets for combating TAM resistance.”

Round 2

Reviewer 2 Report

Comments to authors,

In regard to Comment 2:

Thanks for correcting the company name of Versitech.

If there is a patent that should be included into part 6 of the manuscript.

Additionally, if the authors profit from the sale of the BQ-antibody, that should be noted into the conflict of interest statement at the end of the manuscript.

Author Response

Reviewer 2

Comments to authors,

In regard to Comment 2:

Thanks for correcting the company name of Versitech.

If there is a patent that should be included into part 6 of the manuscript.

Additionally, if the authors profit from the sale of the BQ-antibody, that should be noted into the conflict of interest statement at the end of the manuscript.

Response:

Thank you very much for your suggestions and comments. We sincerely appreciate them. We thought the patent section in this manuscript was whether we employed the findings to apply for a new patent. Therefore, we did not put this information in the original manuscript. Now, we have included this information in the revised manuscript. We have also added a sentence in the main text referring to this

“…and generated a monoclonal antibody that recognizes the unique epitope for the BQ protein (US Patent No. US 10,823,735) [13].

Reviewer 3 Report

1.In Figure 3B, the authors used an indirect method to determine cell viability since the cells for ZR-75 did not form clear colonies at the endpoint. They employed an organic solvent, isopropanol, to extract the pigment from the stained cells and measured the absorbance at 570 nm. They then determined relative cell viability by comparing it with the untreated control. However, the statistical chart does not appear to be appropriate, as it does not present data in a manner that reflects clonogenicity. It is important to note that cell viability and colony formation measure different aspects of cell behavior and cannot be used interchangeably.

2.The IHC image in Figure 5 still has severe issues, as there is very strong cytoplasmic staining and it is difficult to distinguish the nuclear staining. Additionally, it is unclear why they used DAPI, a blue-fluorescent DNA stain, on a light microscope, and there is no visible purple staining.

3.The author ought to provide an explanation for the decision to switch the loading control method for the majority of Western blot experiments and to introduce HSP90 as the new loading control.

4.Thanks for reorganizing the western blot results and detailed explanation. Much better and I would really suggest being more serious about every piece of data presented to readers for your future work.

Author Response

Reviewer 3

Comment 1:

In Figure 3B, the authors used an indirect method to determine cell viability since the cells for ZR-75 did not form clear colonies at the endpoint. They employed an organic solvent, isopropanol, to extract the pigment from the stained cells and measured the absorbance at 570 nm. They then determined relative cell viability by comparing it with the untreated control. However, the statistical chart does not appear to be appropriate, as it does not present data in a manner that reflects clonogenicity. It is important to note that cell viability and colony formation measure different aspects of cell behavior and cannot be used interchangeably.

Response:

Thank you very much for your comments. We agree with the reviewer that the results obtained from pigment extraction reflect cell viability rather than clonogenicity. We have revised the y-axis of the chart and the contents in the methods and figure legend.

In materials and methods:

For the parental cell line MCF-7, since the colonies were easily distinguishable, clonogenicity (%) of MCF-7 shCtrl, MCF-7 shSRSF5.1 and MCF-7 shSRSF5.2 was determined. On the other hand, for the parental ZR-75 cell line, colonies were not easily distinguished, hence the solvent extraction method was applied to determine cell viability of ZR-75 shCtrl, ZR-75 shSRSF5.1 and ZR-75 shSRSF5.2.

In the figure legend:

Clonogenicity (%) was determined in MCF-7 panel, while cell viability (%) was determined in ZR-75 panel due to the lack of a countable colony.

Comment 2

The IHC image in Figure 5 still has severe issues, as there is very strong cytoplasmic staining and it is difficult to distinguish the nuclear staining. Additionally, it is unclear why they used DAPI, a blue-fluorescent DNA stain, on a light microscope, and there is no visible purple staining.

Response:

Thank you for your comments. We apologise for mentioning DAPI as counterstain for viewing on light microscope in our rebuttal. Indeed DAPI is used for immunofluroscent studies. It was a typographic error. We were referring to the chromogen counterstain which stains the nuclei bluish purple. This chromogen DAB/substrate reagent is used routinely for immunohistochemistry on formalin fixed paraffin embedded sections. This is mentioned in Section 2.9 Tissue microarray and immunohistochemistry.

We have re-taken further photomicrographs. For the low nuclear expression of BQ, the bluish purple nuclear stain should be more clearly visible now. Likewise for low nuclear expression of SRSF5, the nuclei are pale against the brownish cytoplasmic staining. In contrast, for the high nuclear expression of both SRSF5 and BQ, the strong nuclear staining comes out clearly in spite of presence cytoplasmic staining. We hope these images are convincing enough. The presence of cytoplasmic stain is inevitable. What is important is that it is the nuclear staining that is scored, not the cytoplasmic stain.

Comment 3

The author ought to provide an explanation for the decision to switch the loading control method for the majority of Western blot experiments and to introduce HSP90 as the new loading control.

Response:

Thank you for your comment. As suggested, some of the results needed to be improved. Therefore, we run the samples again. As an independent validation, we would like to use another loading control HSP90 which is one of the commonly used loading controls. If similar results would be obtained during the revision, this further could further confirm our findings. However, due to time limitation, we could not repeat all experiments. Therefore, we repeat the experiments that might have the potential for quality improvement within this limited period.  

Comment 4:

Thanks for reorganizing the western blot results and detailed explanation. Much better and I would really suggest being more serious about every piece of data presented to readers for your future work.

Response:

Thank you very much for your suggestions and comments during the whole revision process. Indeed, this has helped us improve the manuscript and identify problems during manuscript preparation. We will be more careful in the future. Again, we sincerely thank you for giving us a chance to make corrections and improvements.

Round 3

Reviewer 3 Report

Thank you for your response and for acknowledging the need to improve the results. while HSP90 is used as loading control, I would recommend considering the original loading controls as well to ensure that the results are robust and reliable. The IHC image in Figure 5 still has strong cytoplasmic staining. I have some concerns about the antibody specificity.

Author Response

Report 3:

Thank you for your response and for acknowledging the need to improve the results.

Comment 1:

while HSP90 is used as loading control, I would recommend considering the original loading controls as well to ensure that the results are robust and reliable.

Response:

Thank you for your suggestion. We understand that using GAPDH for the whole manuscript would make it more consistent. One of the reviewers in the previous revision commented the loading control GAPDH and the target protein SRSF5 were not detected on the same blot, and requested that we try to test it on the same blot. GAPDH (mouse origin one) is used routinely for our experiments. Due to the similar molecular weight of SRSF5 and GAPDH, these unfortunately needed to be detected separately. To be able to detect SRSR5 and the loading control on the same blot, we chose one with a larger molecular weight, that is HSP90. The vast difference in molecular weight allows them to be examined on the same blot in order to address the comment from the reviewer. We appreciate the comments from you as well as from the other reviewers. However, we could not satisfy both comments simultaneously. Hence the use of HSP90.  

Comment 2:

The IHC image in Figure 5 still has strong cytoplasmic staining. I have some concerns about the antibody specificity

Response:

Thank you for your comments. BQ is a splice variant of the the NCOR2 gene, NCOR2 being  a nuclear co-repressor that mediates transcription silencing of certain target genes. Both NCOR2 and BQ proteins can be found both in the cytoplasm as well as in the nucleus by IHC, but it is their nuclear expression that reflects their transcription related functions (1-3). It is BQ nuclear expression that modulates tamoxifen response, hence only nuclear expression was assessed. We acknowledge that strong cytoplasmic BQ staining has been an inherent problem. In spite of this, our previous examination of more than 1,000 breast cancer samples by Tissue Microarray (TMA) of both our local cohort and an overseas cohort, had confirmed that nuclear BQ overexpression was significantly associated with tamoxifen resistance, poorer overall and disease-specific survival, and could predict tamoxifen resistance in patients with ER+ve breast cancer with 52.9% sensitivity and 72.0% specificity (2).

References

  1. Tsoi H, Man EPS, Leung MH, Mok KC, Chau KM, Wong LS, et al. KPNA1 regulates nuclear import of NCOR2 splice variant BQ323636.1 to confer tamoxifen resistance in breast cancer. Clinical and Translational Medicine. 2021;11(10).
  2. Gong C, Man EPS, Tsoi H, Lee TKW, Lee P, Ma ST, et al. BQ323636.1, a Novel Splice Variant to NCOR2, as a Predictor for Tamoxifen-Resistant Breast Cancer. Clin Cancer Res. 2018;24(15):3681-91.
  3. Zhang LD, Gong C, Lau SLY, Yang N, Wong OGW, Cheung ANY, et al. SpliceArray Profiling of Breast Cancer Reveals a Novel Variant of NCOR2/SMRT That Is Associated with Tamoxifen Resistance and Control of ER alpha Transcriptional Activity. Cancer Res. 2013;73(1):246-55.